# A noncanonical AR addiction drives enzalutamide resistance in prostate cancer

Yundong He[1,10], Ting Wei[2,10], Zhenqing Ye[2], Jacob J. Orme [3], Dong Lin[4], Haoyue Sheng[1], Ladan Fazli[4], R. Jeffrey Karnes[5], Rafael Jimenez [6], Liguo Wang [2], Liewei Wang[7], Martin E. Gleave[8], Yuzhuo Wang [4,8], Lei Shi[1✉] & Haojie Huang [1,5,9✉]

Resistance to next-generation anti-androgen enzalutamide (ENZ) constitutes a major challenge for the treatment of castration-resistant prostate cancer (CRPC). By performing genome-wide ChIP-seq profiling in ENZ-resistant CRPC cells we identify a set of androgen receptor (AR) binding sites with increased AR binding intensity (ARBS-gained). While ARBS-gained loci lack the canonical androgen response elements (ARE) and pioneer factor FOXA1 binding motifs, they are highly enriched with CpG islands and the binding sites of unmethylated CpG dinucleotide-binding protein CXXC5 and the partner TET2. RNA-seq analysis reveals that both CXXC5 and its regulated genes including *ID1* are upregulated in ENZ-resistant cell lines and these results are further confirmed in patient-derived xenografts (PDXs) and patient specimens. Consistent with the finding that ARBS-gained loci are highly enriched with H3K27ac modification, ENZ-resistant PCa cells, organoids, xenografts and PDXs are hyper-sensitive to NEO2734, a dual inhibitor of BET and CBP/p300 proteins. These results not only reveal a noncanonical AR function in acquisition of ENZ resistance, but also posit a treatment strategy to target this vulnerability in ENZ-resistant CRPC.

[1] Department of Biochemistry and Molecular Biology, Mayo Clinic College of Medicine and Science, Rochester, MN, USA. [2] Division of Biomedical Statistics and Informatics, Mayo Clinic College of Medicine and Science, Rochester, MN, USA. [3] Division of Medical Oncology, Department of Internal Medicine, Mayo Clinic College of Medicine and Science, Rochester, MN, USA. [4] Department of Experimental Therapeutics, BC Cancer Research Centre, Vancouver, BC, Canada. [5] Department of Urology, Mayo Clinic College of Medicine and Science, Rochester, MN, USA. [6] Department of Laboratory Medicine and Pathology, Mayo Clinic College of Medicine and Science, Rochester, MN, USA. [7] Department of Molecular Pharmacology and Experimental Therapeutics, Mayo Clinic College of Medicine and Science, Rochester, MN, USA. [8] The Vancouver Prostate Centre, Department of Urologic Sciences, University of British Columbia, Vancouver, BC, Canada. [9] Mayo Clinic Cancer Center, Mayo Clinic College of Medicine and Science, Rochester, MN, USA. [10]These authors contributed equally: Yundong He, Ting Wei. ✉email: shileihmu@gmail.com; huang.haojie@mayo.edu

The androgen-AR signaling axis is the major driver of AR-positive prostate cancer (PCa)[1–3]. Acting as a transcription factor (TF) either alone or in concert with pioneer factors such as FOXA1, AR operates transcription programs to promote PCa growth and survival by binding to the full or half ARE or a composite motif consisting of ARE and FOXA1 binding element (FOX/ARE)[4,5]. Because of the dependency of most PCa on the AR signaling, androgen deprivation therapy (ADT) has been the mainstay treatment for advanced PCa. Treatment-induced reduction in the level of PSA, which is encoded by the well-studied ARE-driven gene *KLK3*, has served as a reliable surrogate for the effectiveness of ADT. Unfortunately, the disease usually relapses and becomes metastatic CRPC. The importance of AR re-activation in CRPC is reflected in United States Food and Drug Administration approval of next-generation AR signaling inhibitors such as enzalutamide (ENZ) and abiraterone (ABI)[6–8]. While these inhibitors are initially effective, they fail in the majority of CRPC. Different mechanisms related to ENZ and ABI resistance have been identified, which include aberrant glucocorticoid receptor (GR) upregulation[9], AR splice variants (ARVs)[10,11], AR gene mutation[12], somatic acquisition of AR gene enhancers, and AR gene duplication[13–15]. These mechanisms highlight the central role of AR and other nuclear receptors in the development of ENZ and ABI resistance.

A noticeable consensus from various studies is that while most AR-targeted therapy-resistant CRPC continue to express full-length AR (AR-FL)[16–18], expression of ARE-dependent canonical AR target genes such as *KLK3* is often downregulated or completely suppressed[18,19], implying the acquisition of an AR-indifferent phenotype after next-generation AR inhibitor therapy. Additionally, other studies suggest that AR-negative or -low PCa cells or clones pre-exist in untreated primary tumors and they tend to become accentuated in metastatic foci of the CRPC patients, stressing a role of AR-independent mechanisms in therapy resistance and disease progression[20–23]. Considering that a large percentage of CRPC remain AR positive[16,21], in the present study we were interested to define AR-dependent mechanisms of ENZ-resistance. Through genome-wide AR chromatin immunoprecipitation sequencing (ChIP-seq) and RNA sequencing (RNA-seq) analyses in ENZ-resistant (ENZ-R) AR-positive CRPC cells and meta-analysis and immunohistochemistry studies in patient samples, we demonstrated a previously unrecognized noncanonical AR function in ENZ-R CPRC.

## Results

**Gain of AR binding on chromatin in ENZ-resistant CRPC cells.** To recapitulate clinical ENZ resistance, we generated control and ENZ-R AR-positive CRPC variants from C4-2, LNCaP, LAPC4, and VCaP cell lines through long-term (>2 months) treatment of vehicle (DMSO) or ENZ (Supplementary Fig. 1a). Control (C4-2$^{CON}$, LNCaP$^{CON}$, LAPC4$^{CON}$, and VCaP$^{CON}$) and ENZ-R (C4-2$^{ENZ-R}$, LNCaP$^{ENZ-R}$, LAPC4$^{ENZ-R}$, and VCaP$^{ENZ-R}$) cells were maintained by continuous treatment of vehicle and ENZ, respectively. Using C4-2$^{ENZ-R}$ cells as a model, we further confirmed ENZ-R growth of C4-2$^{ENZ-R}$ cells in vivo (Supplementary Fig. 1b). As expected, both RNA-seq and RT-qPCR analyses showed that canonical AR target genes including *KLK3* (PSA), *TMPRSS2*, and *NKX3.1* were transcriptionally downregulated in ENZ-R cells compared to control cells (Supplementary Fig. 1c, d). Little or no expression of ARVs was detectable in ENZ-R C4-2, LNCaP, and LAPC4 cells (Supplementary Fig. 1e, f). The level and nuclear localization of AR-FL protein were comparable between C4-2$^{ENZ-R}$ and C4-2$^{CON}$ cells (Supplementary Fig. 1e, g). Of note, AR knockdown inhibited proliferation of ENZ-R variants of these

cell lines (Supplementary Fig. 2a). These findings suggest that AR-FL is indispensable for ENZ-R growth of CRPC cells.

To determine the molecular mechanism underlying AR-FL-dependent, ARV-independent growth of ENZ-R cells, we performed ChIP-seq for AR and histone H3 lysine 27 (H3K27ac), a histone mark of both active enhancer and transcription start site (TSS) regions[24–26] in C4-2$^{ENZ-R}$ and C4-2$^{CON}$ cells. Among the total AR binding sites (ARBS, 59,780) we identified, 12,652 ARBS were lost (ARBS-lost) in C4-2$^{ENZ-R}$ compared to C4-2$^{CON}$ while 6908 ARBS were gained (ARBS-gained) and 40,220 ARBS were not significantly altered (ARBS-NSA) (Fig. 1a, b). Cis-regulatory element annotation system (CEAS)-based genomic analysis revealed that the frequency of AR occupation at the promoter regions was much higher at ARBS-gained loci compared to ARBS-lost and ARBS-NSA loci (12.2% versus 1.3% and 3.4%, respectively) (Fig. 1c). These results imply a potential role for ARBS-gained in regulating gene transcription and ENZ-R growth of C4-2$^{ENZ-R}$ cells. H3K27ac enrichment at ARBS-gained loci was higher in C4-2$^{ENZ-R}$ cells compared to C4-2$^{CON}$, but opposite results were observed at ARBS-lost and ARBS-NSA loci (Fig. 1d). Because AR has been shown to bind to enhancer regions in many PCa cell systems including both castration-sensitive and -resistant cells, we further compared AR binding and H3K27ac levels at putative enhancers (marked by H3K4me1) in C4-2$^{CON}$ and C4-2$^{ENZ-R}$ cells. A trend of AR binding and H3K27ac level similar to that for total AR binding sites was detected in the ARBS-gained, ARBS-lost, and ARBS-NSA loci located at the putative enhancers (Fig. 1d and Supplementary Fig. 2b). These data suggest that the ARBS-gained sites may be highly transcriptionally active in ENZ-R cells.

**Enrichment of CpG islands and CXXC5 binding at ARBS-gained loci in ENZ-R cells.** TF DNA binding motif analysis revealed that AR and FOXA1 binding motifs were highly enriched at both ARBS-lost and ARBS-NSA, but not at ARBS-gained loci (Fig. 1e), affirming that AR binding at ARBS-gained loci is likely mediated by mechanisms independent of ARE or FOX/ARE elements. In contrast, STEME-based motif enrichment[27] analysis revealed that GC-rich DNA motifs were frequently detected at ARBS-gained loci, but not at ARBS-NSA and ARBS-lost loci (Fig. 1f, g). Approximately 20% (1293 sites) of ARBS-gained loci overlapped CpG island (CpGi) regions, and the frequency of AR binding at the ARBS-gained CpGi loci was much higher than at CpGi loci that overlapped with ARBS-lost or ARBS-NSA loci (Fig. 1h). Notably, ~82% (693 sites) of ARBS-gained loci located in the promoter regions were CpGi-positive (Fig. 1h and Supplementary Table 1). This percentage is much higher than the fraction of all promoters that contain CpGis, which is estimated to be about 50–60%[28]. These data rule out the possibility that the high rate of CpGi enrichment at ARBS-gained loci is simply due to the high chances of ARBS-gained loci at the promoters compared to ARBS-NSA and ARBS-lost loci (Fig. 1c). Meta-analysis of AR binding from a previous report[29] revealed that the percentage of the androgen-independent AR-occupied regions (AI-ORs) located at CpGis was also much higher than that of androgen-dependent AR-occupied regions (AD-ORs) in C4-2B CRPC cells (Fig. 1i). This data further implies the relevance of ARBS-gained and CpGi co-targeted loci in androgen-independent growth of PCa.

CXXC domain proteins selectively recognize unmodified CpG[30,31]. RNA-seq analysis revealed that among the 12 CXXC domain genes in the human genome, only *CXXC4* and *CXXC5* mRNA were substantially upregulated in C4-2 ENZ-R cells although the absolute expression level of *CXXC4* mRNA was very low (Fig. 2a, b and Supplementary Fig. 3a). Similarly, western blot

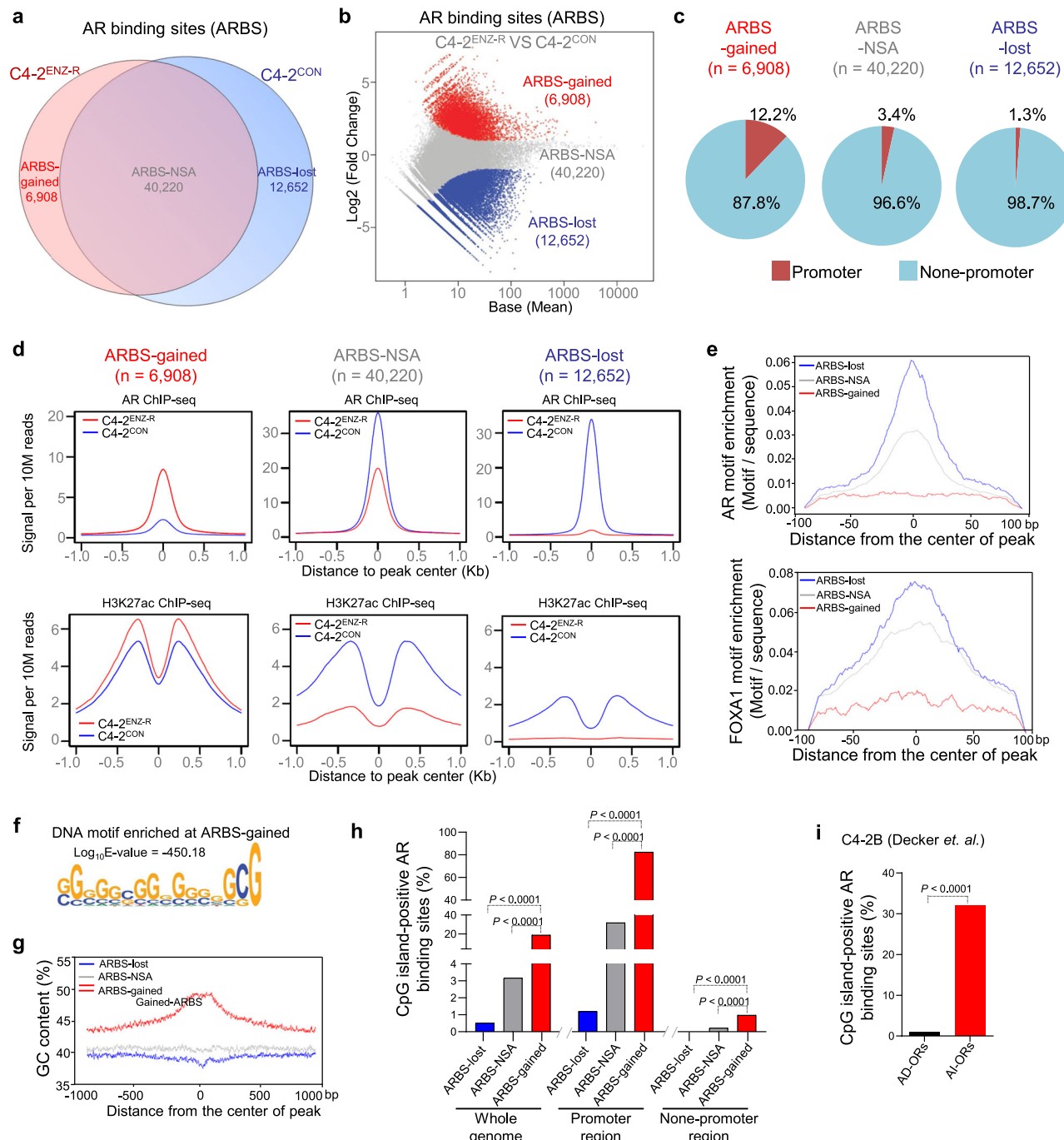

**Fig. 1 Characterization of AR chromatin binding status in ENZ-resistant CRPC cells. a** Venn diagram showing AR chromatin binding sites (ARBS) in C4-2[ENZ-R] and C4-2[CON] cells detected by ChIP-seq. **b** MA plot for AR ChIP-seq. Red, gray, and blue points represent ARBS-gained (logFC > 1, FDR < 0.05), ARBS-NSA (FDR > 0.05), and ARBS-lost loci (logFC < −1, FDR < 0.05) respectively. **c** CEAS genomic analysis of ARBS-lost, ARBS-NSA, and ARBS-gained loci. **d** AR binding signal and H3K27ac modification profiles for ARBS-lost, ARBS-NSA, and ARBS-gained in C4-2[ENZ-R] (red) and C4-2[CON] (blue) cells. **e** Enrichment of AR and FOXA1 DNA binding motif at the ARBS-lost, ARBS-NSA, and ARBS-gained loci in ENZ-R PCa cells. **f** STEME-based DNA motif enrichment analysis of ARBS-gained loci. **g** GC content profiles of ARBS-lost, ARBS-NSA, and ARBS-gained ARBS regions. Each peak was extended 1 kb to up- and down-stream from the center. **h** Percentage of ARBS-lost, ARBS-NSA, and ARBS-gained regions at CpG islands. Statistical significance was determined by two-sided Fisher's exact test. **i** Percentage of the androgen-dependent occupied regions (AD-ORs) and androgen-independent occupied regions (AI-ORs) located at CpG islands in C4-2B cells reported by Decker et al.[29]. Statistical significance was determined by two-sided Fisher's exact test.

analysis showed that CXXC5 protein was markedly upregulated in C4-2[ENZ-R], but CXXC4 protein was hardly detectable in both C4-2[ENZ-R] and C4-2[CON] cells (Fig. 2c and Supplementary Fig. 3b). CXXC5 expression was upregulated by AR knockdown in C4-2[CON] cells, although this regulatory mechanism was lost in ENZ-R cells (Fig. 2c). These data not only indicate a role of AR in

negative regulation of CXXC5 expression in ENZ-sensitive cells, but also provide a plausible explanation for the upregulation of CXXC5 in cells with long-term ENZ treatment. Moreover, CXXC5 level was also much higher in castration-resistant C4-2B cells compared to LNCaP cells from which C4-2B is derived following castration in mice (Supplementary Fig. 3c).

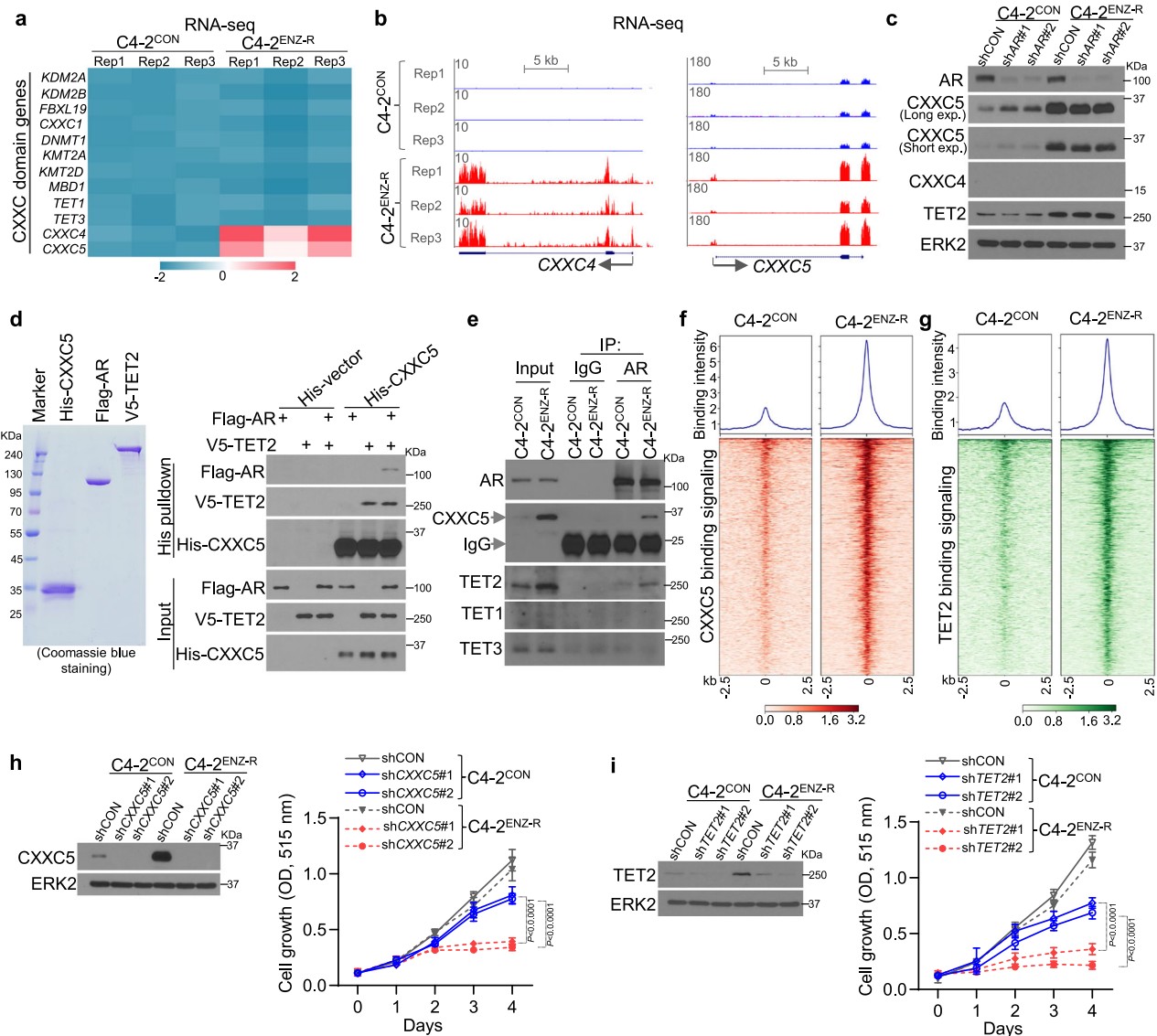

**Fig. 2 CXXC5 and TET2 occupancy at ARBS-gained CpGi-positive sites in ENZ-resistant CRPC cells. a** Heatmap showing RNA-seq read intensity of 12 CXXC domain genes in C4-2^CON and C4-2^ENZ-R cells. **b** UCSC tracks showing RNA-seq signal profiles of CXXC4 and CXXC5 expression in C4-2^CON and C4-2^ENZ-R cells. **c** Western blotting showing the AR, CXXC5, CXXC4, and TET2 protein level in cells expressing control or AR-specific shRNAs. ERK2 was used as a loading control. **d** In vitro pulldown analysis of CXXC5 interaction with AR and TET2 using proteins purified from baculovirus-insect Sf9 cell expression system. **e** Co-IP of endogenous proteins shows AR interaction with CXXC5, TET1, TET2, and TET3 in C4-2^CON and C4-2^ENZ-R cells. **f, g** Heatmap (integration of all replicates) shows ChIP-seq read intensity of CXXC5 and TET2 at ARBS-gained regions in C4-2^CON and C4-2^ENZ-R cells. **h, i** CXXC5 or TET2 was knocked down by specific shRNAs in C4-2^CON and C4-2^ENZ-R cells and the cell proliferation was measured by SRB assay. Data are represented as means ± s.d. ($n = 6$ replicates/group). Statistical significance was determined by two-way ANOVA. Experiments in **c–e** were repeated twice.

TET2 is the only human methylcytosine dioxygenase that lacks the CXXC domain and it functions as a dioxygenase by complexing with CXXC4 or CXX5[32,33]. While forced over-expression of both CXXC4 and CXXC5 resulted in ENZ-R growth of C4-2^CON cells (Supplementary Fig. 3d), only CXXC5 protein was readily detectable and upregulated in C4-2^ENZ-R cells (Fig. 2c and Supplementary Fig. 3b). We therefore focused on CXXC5 in further studies. In vitro protein binding assay showed that CXXC5 physically interacts with TET2, confirming a previous report[32]; however, no CXXC5-AR interaction was detected under similar conditions (Fig. 2d). Intriguingly, CXXC5 can form a protein complex after adding TET2 (Fig. 2d). Co-immunoprecipitation (co-IP) assay showed that AR-CXXC5 interaction was readily detectable in CXXC5-high C4-2^ENZ-R

but not in C4-2^CON cells (Fig. 2e). In contrast, AR interacted with TET2 in both ENZ-R and control cells and their interaction correlated with TET2 protein levels in these cells (Fig. 2e). No AR interaction with TET1 or TET3 was detected in these cell lines (Fig. 2e). These data suggest that increased expression of CXXC5 and TET2 is important for the AR-CXXC5 interaction in ENZ-R cells.

We further examined whether CXXC5 and TET2 play any causal role in ARE-independent recruitment of AR on chromatin in ENZ-R cells. Consistent with our co-IP results (Fig. 2e), CXXC5 and TET2 ChIP-seq studies showed that enrichment of CXXC5 and TET2 at ARBS-gained-CpGi sites was much higher in C4-2^ENZ-R cells compared to control cells (Fig. 2f, g). Importantly, knockdown of CXXC5 or TET2 significantly

inhibited C4-2$^{ENZ-R}$ cell proliferation (Fig. 2h, i), indicating that CXXC5 and TET2 play important roles in ENZ-R growth of PCa cells.

**Upregulation of cell lineage and cancer-promoting genes located at ARBS-gained CpGi loci.** To define the potential downstream effectors of enhanced AR binding at ARBS-gained CpGi sites in ENZ-R cells, we performed integration analysis of AR ChIP-seq and RNA-seq data in C4-2$^{CON}$ and C4-2$^{ENZ-R}$ cells. We found that 226 ARBS-gained CpGi loci-associated genes were significantly upregulated in C4-2$^{ENZ-R}$ cells compared to control cells (Fig. 3a; Supplementary Data 1). Gene set enrichment analysis (GSEA) showed that genes involved in cell lineage development and organ morphogenesis were significantly upregulated (Fig. 3b). These results are consistent with changes in cell morphology and migration in C4-2$^{ENZ-R}$ cells (Supplementary Fig. 4a, b).

Specifically, we found that genes involved in regulation of cell lineage transition and tumor progression including inhibitor of differentiation 1 (*ID1*), actin cytoskeletal regulator *PFN2* and *ID3*[34–39] are among the top of the upregulated genes in ENZ-R cells compared to control cells (Fig. 3a, c; Supplementary Data 1). ChIP-seq data showed that the levels of AR, CXXC5, and TET2 protein occupancy at the promoters and/or potential enhancers of these genes were much higher in C4-2$^{ENZ-R}$ cells than those in control cells, but it was not always the case for FOXA1 binding at these loci (Fig. 3c). H3K27ac levels at these loci were also higher in C4-2$^{ENZ-R}$ cells relative to control cells (Fig. 3c), consistent with active transcription of *ID1*, *PFN2*, and *ID3* genes in ENZ-R cells. This data is also concordant with robust downregulation of ID1 repressor THBS[39] and upregulation of ID1 activator MMP14[40] in C4-2$^{ENZ-R}$ cells (Supplementary Fig. 4c). In contrast, expression of canonical AR target genes (e.g. *KLK3*, *TMPRSS2*, and *NKX3.1*) as well as AR and FOXA1 binding and H3K27ac levels at these gene loci were dramatically downregulated in C4-2$^{ENZ-R}$ cells, and no obvious CXXC5 or TET2 binding peaks were detected at these loci (Fig. 3c).

To determine whether these noncanonical AR genes are relevant to resistance of AR signaling inhibitor therapy in CRPC patients, we performed meta-analysis of RNA-seq data in the SU2C database[41]. We found that high expression of CXXC5 associated with poor overall survival of CRPC patients treated with AR signaling inhibitors (including ENZ and ABI) although the *P* value of the association was slightly greater than 0.05 (Fig. 3d). This could be explained in the context that the impact of CXXC5 on therapy resistance may rely greatly on the functions of its downstream effectors. Indeed, high expression scores of CXXC5-regulated noncanonical AR signature genes (*n* = 226), including *ID1* and *PFN2* (Fig. 3a; Supplementary Data 1), significantly associated with unfavorable outcome of AR signaling inhibitor therapy in CRPC patients (Fig. 3e–g). Particularly, high expression of *ID1* or *PFN2* gene alone also significantly associated with poor outcome of AR signaling inhibitor therapy in these patients whereas expression of *KLK3* or *TMPRSS2* correlated with favorable outcome (Fig. 3h). CXXC4 was excluded from analysis since its expression was much lower compared to CXXC5 level in these patients (Supplementary Fig. 4d). Together, these data support a model in which working through TET2 and AR, increased expression of CXXC5 induces expression of noncanonical AR targets such as ID1 and PFN2, thereby contributing to ENZ-R progression in CRPC.

**Importance of noncanonical AR target genes in ENZ-resistant growth of CRPC cells.** We further examined the functional relevance of noncanonical AR target genes in ENZ resistance.

Western blot and RT-qPCR analyses confirmed robust upregulation of the noncanonical AR targets ID1, PFN2, and ID3 at both mRNA and protein levels in ENZ-R C4-2 cells compared to control cells (Fig. 4a, b). In contrast, FOXA1 expression was moderately decreased in C4-2$^{ENZ-R}$ cells (Fig. 4a), consistent with the decreased FOXA1 binding in the canonical AR target genes such as *KLK3, TMPRSS2*, and *NKX3.1* (Fig. 3c). Strikingly, expression of these noncanonical AR genes remains AR-dependent in C4-2$^{ENZ-R}$ cells (Fig. 4c, d). Similar to the effect of AR, knockdown of CXXC5 decreased expression of these proteins in ENZ-R cells (Fig. 4d). In contrast, expression of canonical AR target genes such as *KLK3, TMPRSS2*, and *NKX3.1* was only moderately reduced by AR knockdown in C4-2$^{ENZ-R}$ cells (Fig. 4c), and this is presumably because the basal expression level of these genes has already been very low in C4-2$^{ENZ-R}$ cells relative to control cells (Figs. 3c and 4c).

We found that proliferation of C4-2$^{ENZ-R}$ cells was more reliant on noncanonical AR targets ID1, PFN2, and ID3 compared to control cells (Fig. 4e–g), supporting the importance of these genes in ENZ-R growth of C4-2 cells. Similarly, AR expression was unchanged in three other ENZ-R CRPC cell lines; however, expression of CXXC5, ID1, PFN2, and ID3 were robustly upregulated in ENZ-R cells compared to control cells except TET2 expression in VCaP$^{ENZ-R}$ cells (Supplementary Fig. 4e). We further showed that growth of these ENZ-R cell lines remained AR-dependent to the extent similar to the control cells although these cells relied more on CXXC5 or ID1 for proliferation than control cells (Supplementary Fig. 4f). These data indicate that CXXC5-dependent expression of noncanonical AR target genes is important for ENZ-R growth of CRPC cells.

Similar to the effect of CXXC5, knockdown of TET2 not only abolished expression of the noncanonical AR targets at both mRNA and protein levels, but also decreased AR occupancy at these gene promoters (Fig. 4d, h, i). Importantly, gene knockdown and rescue experiments showed that the effects of TET2 on expression of noncanonical AR target genes and growth of C4-2 ENZ-R cells were independent of the catalytic activity of TET2 (Fig. 4j–n). Moreover, restored expression of CXXC5-WT, but not the TET2 binding-deficient mutant CXXC5(1–250) rescued CXXC5 knockdown-induced downregulation of these genes and inhibition of C4-2 ENZ-R cell growth (Supplementary Fig. 5a–d). These results indicate that the binding of CXXC5 with TET2 is critical for CXXC5-mediated ENZ resistance. Furthermore, we performed methylated DNA immunoprecipitation (MedIP) assay and demonstrated that the DNA 5mC and 5hmC levels at these noncanonical AR target loci were hardly detectable compared to the positive control locus and were unaffected by either knockdown of TET2 or elimination of TET2 catalytic activity (Supplementary Fig. 5e). These results provide further support to the model in which CXXC5 directly binds to unmethylated CpGs and subsequently recruits TET2 and AR to the noncanonical AR target gene loci, thereby leading to ARE-independent, AR-dependent transcription of noncanonical AR target genes and ENZ-R growth of CRPC cells.

**NEO2734, an oral dual inhibitor of BET and CBP/p300 proteins overcomes ENZ resistance.** To define strategies to overcome ENZ resistance in CRPC, we surveyed the sensitivity of C4-2$^{ENZ-R}$ cells to inhibitors targeting an array of functionally diversified signaling pathways. By comparing the half maximal inhibitory concentration (IC50) of these inhibitors between C4-2$^{ENZ-R}$ and control cells, we found that C4-2$^{ENZ-R}$ cells were highly sensitive to the CBP/p300 inhibitor CPI637[42] (Fig. 5a; Supplementary Fig. 6a, b). This result is consistent with the observation that H3K27ac level was much higher at ARBS-gained

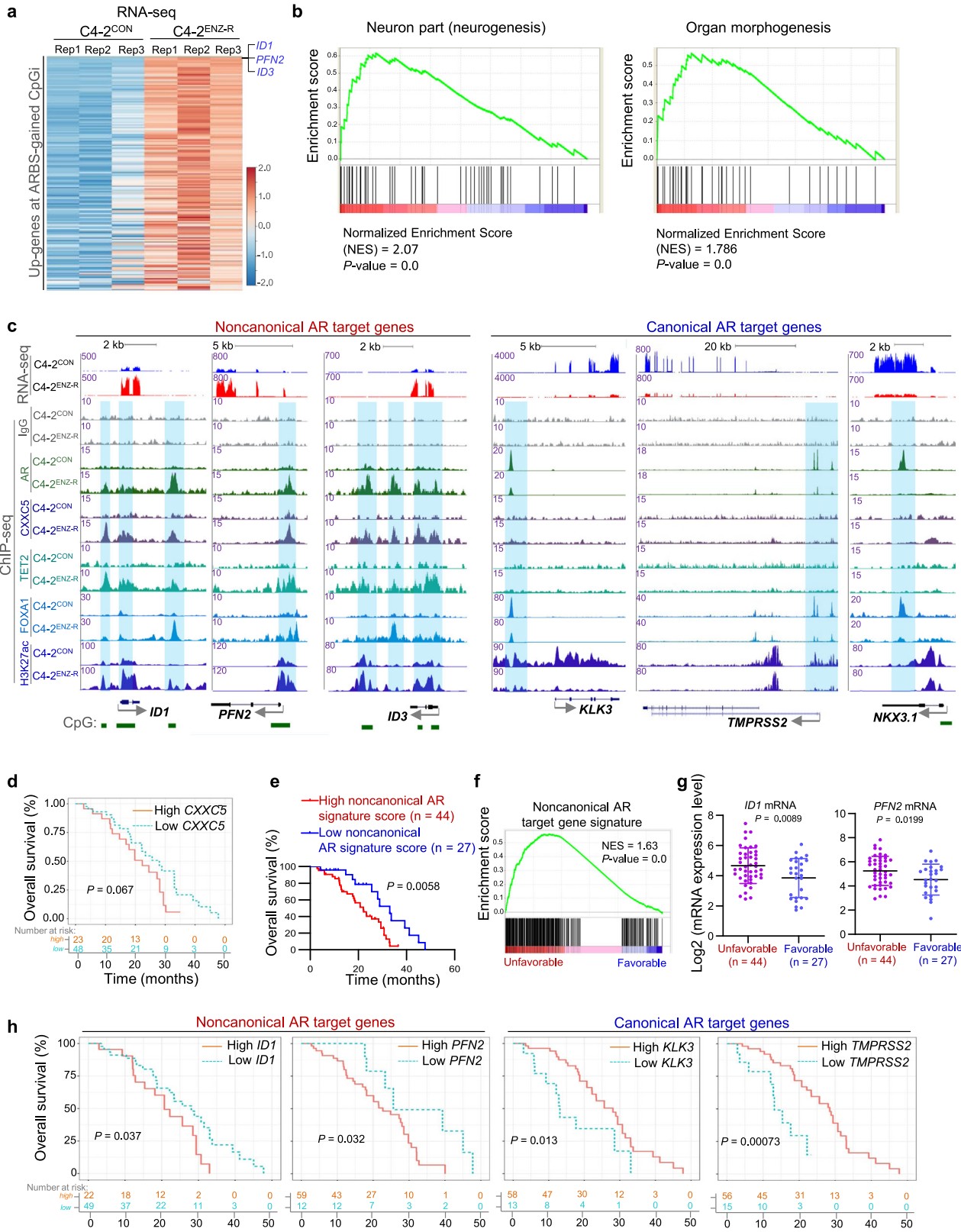

loci in ENZ-R cells compared with control cells (Figs. 1d and 3c). Similar to CRPC cells reported previously[43], C4-2[ENZ-R] cells were highly sensitive to the BET inhibitor JQ1 (Fig. 5a; Supplementary Fig. 6a, b). Most importantly, the combination of CBP/p300 and BET inhibitors completely blocked C4-2[ENZ-R] cell growth in vitro and in vivo (Supplementary Fig. 6c, d), providing a strong

rationale to investigate the therapeutic efficacy of NEO1132 and NEO2734, two dual inhibitors of BET and CBP/p300 proteins[44] in ENZ-R CRPC cells. We found that C4-2[ENZ-R] cells were sensitive to both NEO1132 and NEO2734 and that the efficacy of NEO2734 was much greater at lower doses (Fig. 5b). We therefore chose to test the effect of NEO2734 in other ENZ-R CRPC

**Fig. 3 Identification of upregulated nonconanical AR target genes in ENZ-resistant CRPC cells. a** Heatmap shows RNA-seq read intensity of the upregulated genes (see the list of genes in Supplementary Data 1) at the ARBS-gained-CpGi loci in C4-2$^{ENZ-R}$ cells. **b** GSEA analysis of upregulated signature genes associated with the ARBS-gained-CpGi sites in ENZ-resistant CRPC cells. **c** UCSC tracks (integration of all replicates) show profiles of RNA-seq signals, ChIP-seq signals of IgG, AR, CXXC5, TET2, FOXA1, and H3K27ac at ARBS-gained-CpGi gene loci (*ID1, PFN2, ID3*) or ARBS-lost gene loci (*KLK3, TMPRSS2*, and *NKX3.1*). **d** Kaplan–Meier survival analyses of the association of *CXXC5* expression with the overall survival of SU2C CRPC patients treated with AR signaling inhibitors (including ENZ and ABI). For overall survival analysis, log-rank (Mantel–Cox) test was used to determine the statistical difference between stratified groups. **e** Kaplan–Meier survival analyses of the association of noncanonical AR gene expression score (see Methods) with the overall survival of SU2C CRPC patients treated with AR signaling inhibitors (including ENZ and ABI). For overall survival analysis, log-rank (Mantel–Cox) test was used to determine the statistical difference between stratified groups. **f** GSEA plot showing the enrichment of noncanonical AR genes in the indicated groups with favorable or unfavorable outcome of SU2C CRPC patients treated with AR signaling inhibitors (including ENZ and ABI). 44 patients unfavorable versus 27 favorable patients; Statistical significance was determined by two-sided Kolmogorov–Smirnov test. **g** Expression of *ID1* and *PFN2* genes in the indicated groups with favorable ($n = 27$) or unfavorable ($n = 44$) outcome of SU2C CRPC patients treated with AR signaling inhibitors (including ENZ and ABI). Data shown as means ± s.d. Statistical significance was determined by unpaired two-tailed Student's *t* tests. **h** Kaplan–Meier survival analyses of the association of noncanonical AR gene (*ID1* and *PFN2*) and canonical AR gene (*KLK3* and *TMPRSS2*) expression with the overall survival of SU2C CRPC patients treated with AR signaling inhibitors (including ENZ and ABI). For overall survival analysis, log-rank (Mantel–Cox) test was used to determine the statistical difference between stratified groups.

cell lines LNCaP$^{ENZ-R}$, VCaP$^{ENZ-R}$ and LAPC4$^{ENZ-R}$ and similar results were obtained (Fig. 5c).

NEO2734 treatment suppressed expression of AR and CXXC5 proteins as well as their targets ID1, PFN2, and ID3 in a dose-dependent manner, but had no effect on TET2 protein expression (Fig. 5d). ChIP-seq data showed that the H3K27ac levels at the loci of *AR* and *CXXC5* and their target genes *ID1, PFN2*, and *ID3* were higher in C4-2 ENZ-R cells compared to control cells (Supplementary Fig. 6e). These results were further confirmed by ChIP-qPCR, and similar results were obtained for BRD4 and p300 binding at these loci (Supplementary Fig. 6f). However, NEO2734 treatment not only decreased H3K27ac level and BRD4 and p300 binding at *ID1, PFN2*, and *ID3* gene loci, but also inhibited the expression of these genes (Supplementary Fig. 6g, h). Furthermore, consistent with inhibition of CXXC5 and AR expression by NEO2734 (Fig. 5d; Supplementary Fig. 5i), this dual inhibitor also decreased occupancy of CXXC5, TET2, and AR proteins at the *ID1, PFN2*, and *ID3* gene loci (Supplementary Fig. 5j); however, restored expression of CXXC5 via ectopic transfection completely reversed NEO2734-induced inhibition of the occupancy of CXXC5, TET2, and AR (only partially) at the promoters of these noncanonical AR target genes, expression of these genes and growth of C4-2 ENZ-R CRPC cells (Supplementary Fig. 6i–l). These data support the notion that CXXC5-dependent noncanonical AR signaling is a viable therapeutic target of the BET-CBP/p300 dual inhibitor in ENZ-R CRPC cells.

The in vitro findings prompted us to assess the anticancer efficacy of NEO2734 in ENZ-R CRPC tumors in vivo. While C4-2$^{ENZ-R}$ tumors were resistant to ENZ treatment, their growth was substantially inhibited by CPI637 or JQ1 alone (Fig. 5e–g). Most importantly, administration of NEO2734 abrogated C4-2$^{ENZ-R}$ xenograft growth in mice with no obvious effect on body weight (Fig. 5e–h). IHC analysis showed that expression of AR, CXXC5, ID1, PFN2, and Ki67 proteins was markedly decreased in tumors treated with NEO2734, but not ENZ. In contrast, cleaved-caspase3 level was not significantly affected (Fig. 5i). These results indicate that dual inhibition of BET and CBP/p300 family proteins impairs noncanonical AR target gene expression and the growth of ENZ-R CRPC tumors in vivo.

**Evaluation of the noncanonical AR activity in ENZ-R clinical specimens.** To further validate the clinical relevance of non-canonical AR activity in the development of ENZ resistance, we performed IHC to examine protein expression of the AR-CXXC5-ID1 axis in a group of patients diagnosed with hormone naive PCa ($n = 24$), CRPC ($n = 16$), or ENZ-R PCa ($n = 13$). We found that AR protein was readily expressed in all specimens

except one ENZ-R and two CRPC cases (Fig. 6a, b; Supplementary Fig. 7a; Supplementary Data 2). While AR protein level was slightly lower in ENZ-R specimens than hormone naive PCa, expression of CXXC5 and its downstream target ID1 was significantly upregulated in ENZ-R CRPC (Fig. 6a, b; Supplementary Data 2), consistent with our findings in vitro (Fig. 4a; Supplementary Fig. 3c). Similarly, CXXC5 protein was upregulated in most ENZ-untreated CRPC patient specimens (Fig. 6a, b). Intriguingly, strong staining of CXXC5 and ID1 protein was also detected in two ABI-resistant CRPC samples we examined (Supplementary Fig. 7b; Supplementary Data 2), suggesting the upregulation of noncanonical AR activity could also confer resistance to ABI.

Next, we sought to therapeutically inhibit noncanonical AR activity in ENZ-R CRPC in clinically relevant models. To this end, we evaluated the efficacy of the dual inhibitor NEO2734 in patient-derived xenograft (PDX) models. We have generated and maintained CRPC and ENZ-R PDX models as previously published[45]. As expected, CRPC PDX tumors responded to ENZ treatment but ENZ-R PDXs did not (Supplementary Fig. 8a). Western blot analysis showed that expression of CXXC5 and TET2 and their downstream targets ID1, PFN2, and ID3 was higher in ENZ-R PDXs compared to CRPC controls (Fig. 6c). These data indicate that the ENZ-R PDX is a suitable model for further study.

We first tested NEO2734 in organoids established from ENZ-R and CRPC PDXs. We found that CRPC organoids were sensitive to both ENZ and NEO2734 treatment, while ENZ-R organoids were highly sensitive to NEO2734 but resistant to ENZ up to a very high dose (Fig. 6d, e; Supplementary Fig. 8b). To explore further, we treated ENZ-R PDX tumors in castrated male mice with CPI637, JQ1, or NEO2734 individually. ENZ-R PDXs responded well to CPI637, JQ1, or NEO2734 but not to ENZ (Fig. 6f–h) and most importantly, NEO2734 outperformed CPI637 or JQ1 in tumor suppression (Fig. 6f–h). In agreement with these observations, NEO2734 significantly diminished the levels of AR, CXXC5, ID1, and Ki67 protein in ENZ-R PDXs but had no discernable effect on expression of cleaved Caspase-3 (Supplementary Fig. 8c). These data indicate that NEO2734 overcomes the aberrant noncanonical AR activity and ENZ resistance in AR-positive CRPC.

## Discussion

Increasing evidence indicates that the number of AR-negative/low CRPC variants such as neuroendocrine PCa (NEPC) and double-negative PCa (DNPC) substantially increases after treatment of the next-generation AR signaling inhibitors including ENZ and

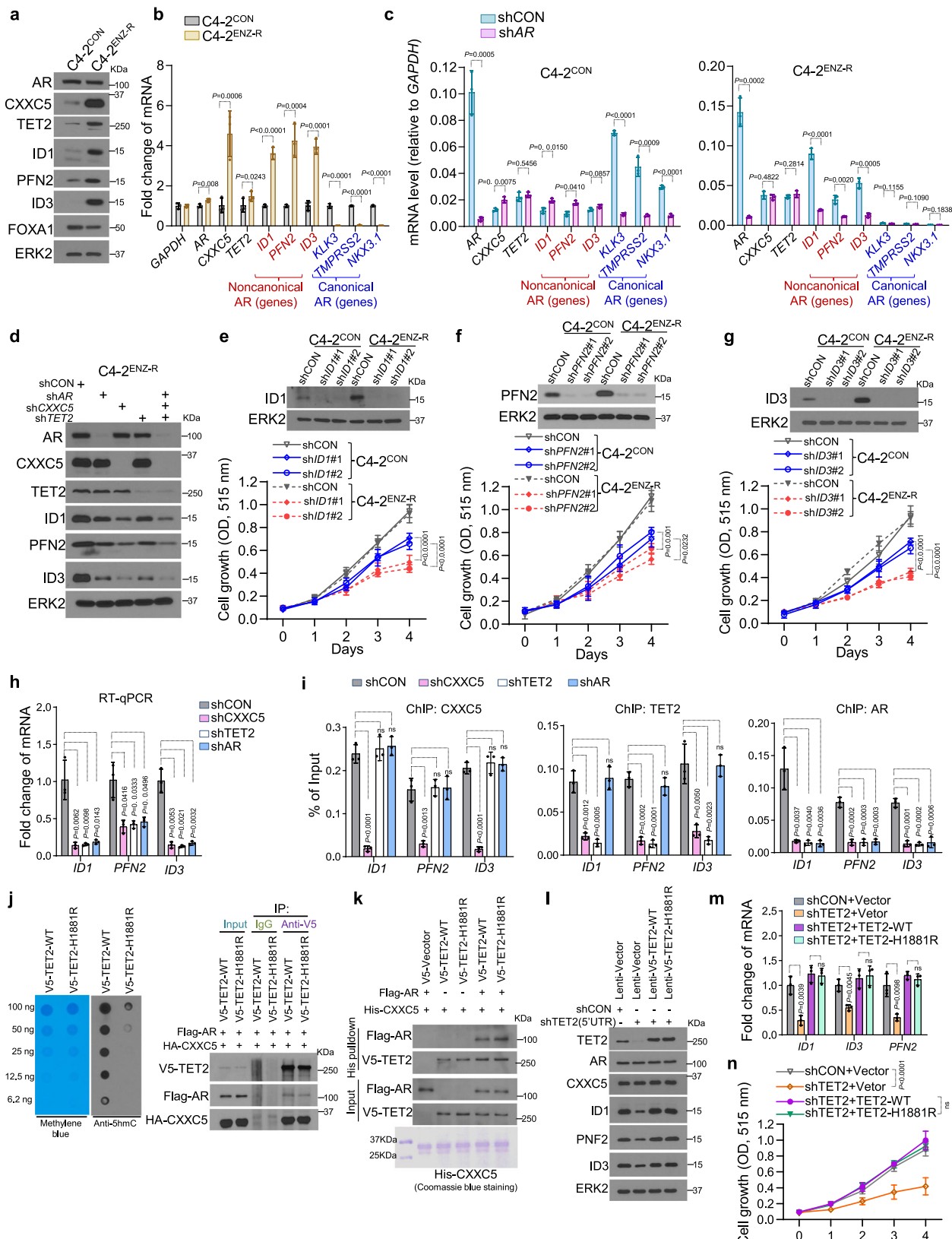

ABI[16,19,46–49]. These studies further show that several AR-independent mechanisms (Fig. 7), including FGF signaling, N-Myc overexpression and upregulation of master transcription regulators BRN2 and ONECUT2, play important roles in lineage plasticity and AR-targeted therapy resistance in CRPC. Notably, the AR-FL is well expressed in the majority of ENZ- and

ABI-resistant CRPC tumors although they have lost expression of canonical ARE-dependent AR target genes such as KLK3, implying that these PSA-negative CRPC are AR-indifferent tumors[16,18,19]. In the current study we found that although expression of canonical AR target genes such as KLK3, TMPRSS2, and NKX3.1 was hardly detectable in AR-FL-expressing ENZ-R

**Fig. 4 The importance of noncanonical AR-targeted genes in ENZ-resistant growth of CRPC cells. a** Western blot showing the indicated proteins expression in C4-2[CON] and C4-2[ENZ-R] cell lines. RT-qPCR analysis shows the indicated genes expression in C4-2[CON] and C4-2[ENZ-R] cells (**b**) or in control and AR knockdown C4-2[ENZ-R] cells (**c**). Data shown means ± s.d., (*n* = 3 replicates/group). **d** Western blot analysis of the indicated proteins in C4-2[ENZ-R] cells infected with the indicated lentivirus expressing gene-specific shRNAs for 96 h. **e–g** Cell proliferation analysis of C4-2[CON] and C4-2[ENZ-R] cells infected by the specific shRNAs. Data shown means ± s.d., (*n* = 6 replicates/group). **h** RT-qPCR analysis of expression of *ID1*, *PFN2*, and *ID3* in C4-2[ENZ-R] cells transfected control or AR, CXXC5, or TET2-specific shRNAs. Data shown as means ± s.d. (*n* = 3 replicates/group). **i** ChIP-qPCR analysis of CXXC5, TET2, and AR occupancy at genomic loci of *ID1*, *PFN2*, and *ID3* in C4-2[ENZ-R] cells transfected control or AR, CXXC5, or TET2-specific shRNAs. Data shown as means ± s.d. (*n* = 3 replicates/group). **j** Genomic DNA purified from 293T cells overexpressing wild-type (WT) or mutant TET2 (H1881R) was stained with methylene blue or blotted using anti-5hmC (left). Co-IP analysis of interaction of TET2-WT and TET2-H1881R with CXXC5 and AR in 293T cells (right). **k** His-tag pulldown analysis of His-tagged CXXC5 interaction with AR and TET2 proteins produced by in vitro transcription/translation. **l** Western blot showing the indicated proteins expression in C4-2[ENZ-R] cells infected with lentivirus expressing indicated shRNAs or expression vectors. **m** RT-qPCR analysis of expression of *ID1*, *PFN2*, and *ID3* in C4-2[ENZ-R] cells infected with lentivirus as in **l**. Data shown as means ± s.d. (*n* = 3 replicates/group). **n** Cell proliferation analysis of C4-2[ENZ-R] cells with knockdown of endogenous TET2 and restored expression of TET2-WT or TET2-H1881R. Data shown as means ± s.d. (*n* = 6 replicates/group). Statistical significance was determined by unpaired two-tailed Student's *t* tests in **b**, **c**, **h**, **i**, **m** and by two-way ANOVA in **e**, **f**, **g**, **n**. Experiments in **a**, **d**, **j**, **k** were repeated twice.

CRPC cells, depletion of AR inhibited the growth of these cells, stressing that AR-FL remains functionally important rather than indifferent in the growth of ENZ-R CRPC. In support of this hypothesis, we identify a subset of ARE-independent non-canonical AR target genes including *ID1*, *PFN2*, and *ID3* in ENZ-resistance cells. Most importantly, we demonstrate that expression of these genes is not only important for ENZ-R growth of CRPC cells in vitro and in vivo, but also associates with poor overall survival of ENZ-treated CRPC patients, although one limitation of our current study is the small sample size of ENZ-R CRPC and the significance of our findings can be improved by the validation studies in larger cohorts. Nevertheless, our study uncovers a previously uncharacterized ARE-independent, but AR-dependent mechanism responsible for ENZ resistance in CRPC (Fig. 7). These findings also reveal a paradigm whereby ENZ-R variants might be "indifferent" to canonical ARE-dependent AR activity but still relies on the noncanonical AR-dependent function for growth and progression (Fig. 7).

A unique feature shared by the noncanonical AR target genes we identified in ENZ-R cells is the paucity of ARE and FOX/ARE elements. Instead, these loci are located in the GC-rich promoter and non-promoter regions with significant overlap with the CpGis. In agreement with these observations, our genome-wide RNA-seq and ChIP-seq analyses show that CpG dinucleotide-binding protein CXXC5 is largely upregulated in ENZ-R CRPC cells and that CXXC5 and its binding partner TET2 occupy at the noncanonical AR target gene loci. We further demonstrate that TET2 interacts with AR and that AR is indirectly recruited into the CpGi-rich regions by CXXC5 via its interaction with TET2 (Fig. 7). These findings provide a mechanistic explanation as to why AR occupies at these noncanonical target loci lacking ARE or FOX/ARE motifs. In agreement with the observation that AR is recruited into these loci through an indirect mechanism, AR binding intensities are lower at the noncanonical ARE-null loci compared to ARE-positive loci. These findings also mechanistically explain why the AR-mediated, ARE-independent transcriptional program is insensitive to ENZ treatment (Fig. 7).

Through unbiased signaling pathway inhibitor survey, we demonstrate that ENZ-R cells are hyper-sensitive to the dual inhibitors of BET and CBP/p300 proteins such as NEO2734[50–52]. In agreement with these results, we show that chromatin occupancy of CXXC5 and H3K27ac levels are positively correlated at noncanonical AR target loci and that the BET-CBP/p300 dual inhibitor treatment inhibits CXXC5 protein expression and decreases BRD4 and p300 binding and H3K27ac levels at these loci. These mechanistic bases provide a strong rationale to clinically investigate the anticancer efficacy of bromodomain inhibitors and BET-CBP/p300 dual inhibitors in ENZ-R CRPC

which are addicted to the noncanonical AR activity for survival (Fig. 7). Furthermore, we show that CXXC5 and its downstream effector ID1 are also upregulated in a subset of CRPC variants relapsed from ADT or ABI treatment. These data suggest that the molecular mechanism and targeting strategy uncovered in ENZ-R CRPC may be applicable to other types of therapy-resistant CRPC and further investigation is warranted.

## Methods

**ChIP, ChIP-seq, and bioinformatics analyses.** For ChIP experiment, chromatin was cross-linked for 15 min at room temperature by adding 11% formaldehyde/PBS solution in cell culture medium[53]. Cross-linked chromatin was then sonicated, diluted, and immunoprecipitated with Protein G-plus Agarose beads (Bio-Rad®) prebound with antibodies at 4 °C overnight. Immunoprecipitation antibodies were AR (2 µg/sample; #sc-816, Santa Cruz Biotechnology), FOXA1 (2 µg/sample; #ab23738, Abcam), H3K27ac (2 µg/sample; #ab4729, Abcam), CXXC5 (2 µg/sample; #16513-1-AP, Proteintech), TET2 (2 µg/sample; #ab94580, Abcam) and Rabbit IgG (2 µg/sample; #ab171870, Abcam). Precipitated protein-DNA complexes were eluted and cross-linking was reversed at 65 °C for 12 h. ChIP-seq libraries were prepared using previously described methods[54]. High-throughput sequencing (51 nt, pair-end) was performed using the Illumina HiSeq[TM]4000 platforms at the Mayo Clinic Genome Core Facility. All short reads were mapped to the human reference genome (GRCh37/hg19) using bowtie2 (version 2.1.0) with default configurations[55]. Reads mapped to multiple positions greater than 2 were discarded, and the remaining reads were then used for peak calling by MACS2 (version 2.0.10) with a *p* value cutoff of 1e-5 (macs2 callpeak -bdg -SPMR -f BAM -p 1e-5)[56]. Peaks located on curated blacklists such as centromere regions were removed (https://sites.google.com/site/anshulkundaje/projects/blacklists). ChIP-seq tag intensity tracks (bedGraph files) were generated by MACS2, and converted into bigWig files using UCSC "wigToBigWig" tool. Genomic distribution of peaks with regard to TSS, and the association of peaks to target genes was performed by Homer package[57,58]. The de novo-motif discovery was conducted using STEME, an efficient EM algorithm to find motifs in large data sets[27,59]. Heat maps were drawn by deepTools 2.0, and customized python scripts were used for other analyses as needed.

**RNA-seq analyses and real-time PCR.** For RNA-seq, libraries were prepared using Illumina TruSeq RNA prep kit and standard protocol. The RNA libraries were sequenced as 51 nt pair-end reads at one sample per lane of an Illumina HiSeq 2500, generating an average of 265 million reads per sample. All reads were aligned to the human reference genome (GRCh37/hg19) by TopHat 2.0.9 using default options. Gene expression counts were generated using RSeQC and expression differential analysis was conducted using edgeR (version 3.6.8)[60]. GSEAs were carried out using the signature scores per gene (fold change) in pre-ranked mode with default settings[61]. Gene expression was determined by real-time quantitative PCR (qPCR) using Power SYBR Green (4368708, Thermo Fisher Scientific). Primer sequences used for qPCR are listed in Supplementary Data 3.

**Noncanonical AR target gene expression score.** The expression values (log2 (FPKM)) of noncanonical AR target genes (Supplementary Data 1) were converted to Z-scores by $Z = (x - \mu)/\sigma$[47], where x is the raw log2(FPKM), µ is the mean and σ is the standard deviation across all samples of a gene. Finally, the Z-scores were summed across all genes to represent the expression score of noncanonical AR target genes. The RNA-seq and clinic data of metastatic CRPC (SU2C/PCF Dream Team)[41], treated with ENZ and ABI, was derived from cBioPortal (http://www.cbioportal.org/)[62].

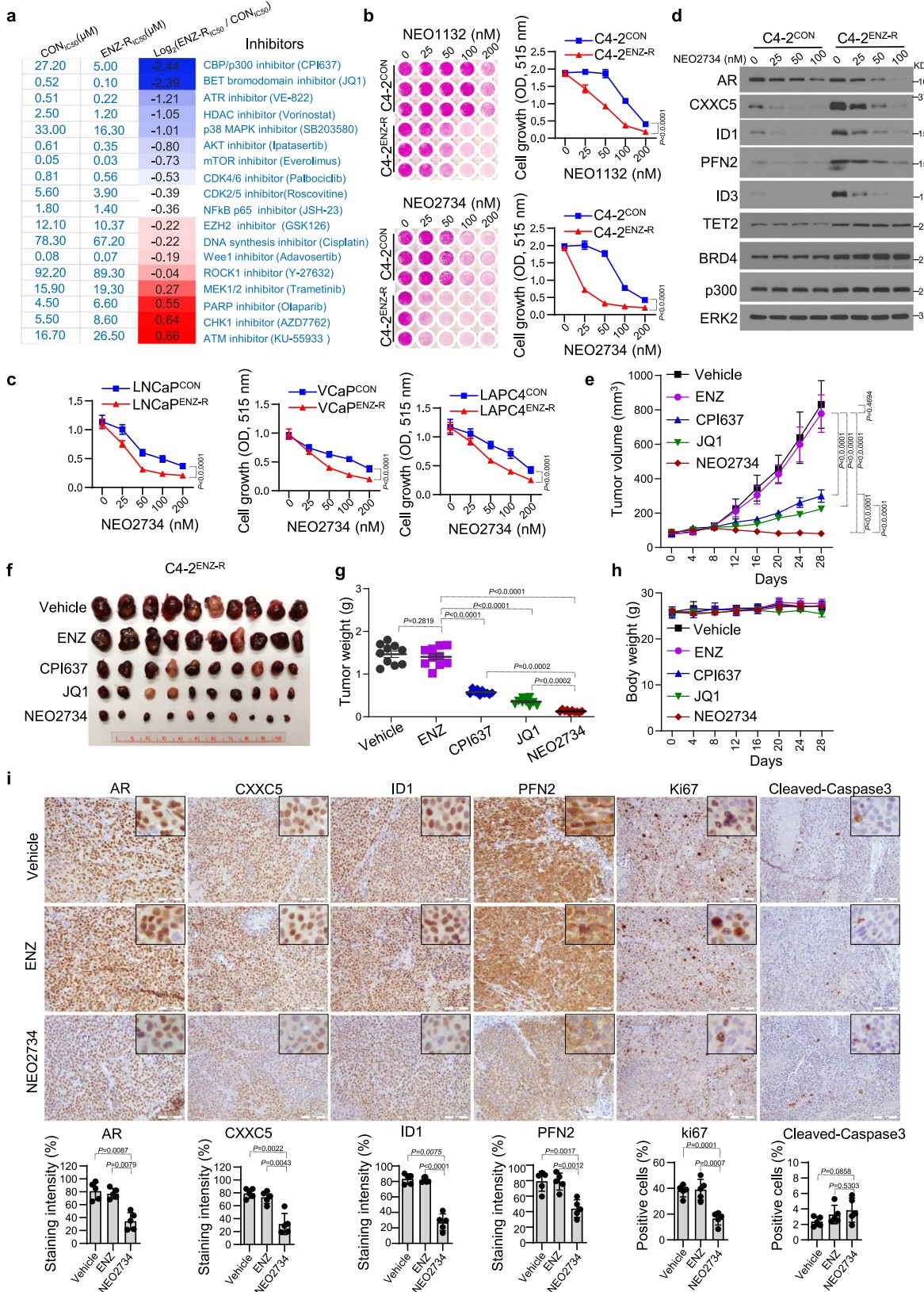

**Cell lines and cell culture**. LNCaP, VCaP, and LAPC4 PCa cell lines and 293T cell line were purchased from ATCC. C4-2 cells were purchased from Uro Corporation (Oklahoma City, OK). C4-2, LNCaP, VCaP, and LAPC4 cells were maintained at 37 °C and 5% CO2 in RPMI 1640 containing 10% fetal bovine serum (FBS) and 1% antibiotic/antimycotic (Thermo Fisher Scientific). 293T cells were maintained in DMEM medium with 10% FBS. To establish ENZ resistant cell lines, C4-2, LNCaP,

VCaP, and LAPC4 cells were cultured in medium containing ENZ. Concentrations of ENZ were gradually increased to 30 μM for C4-2, VCaP, and LAPC4 cell lines except that LNCaP was up to 5 μM ENZ, while control cell lines were cultured in medium with the same amount of vehicle without ENZ. Mycoplasma contamination was tested by the PCR Mycoplasma Detection Set (Takara, Otsu, Japan). All cell lines are negative for mycoplasma contamination.

**Fig. 5 ENZ-resistant CRPC cells are sensitive to BET and CBP/p300 inhibitors in vitro and in vivo. a** Heatmap shows the sensitivity of C4-2$^{CON}$ and C4-2$^{ENZ-R}$ cells to the indicated pathway inhibitors. **b** Cell proliferation analysis shows the inhibitory effect of BET-CPB/p300 dual inhibitors NEO1132 and NEO2734 on C4-2$^{CON}$ and C4-2$^{ENZ-R}$ cell proliferation. Data shown as means ± s.d. ($n = 3$ replicates/group). Statistical significance was determined by two-way ANOVA. **c** Cell proliferation analysis of control and ENZ-R LNCaP, VCaP, and LAPC4 cells treated with the indicated concentration of NEO2734. Data shown as means ± s.d. ($n = 6$ replicates/group). Statistical significance was determined by two-way ANOVA. **d** Western blot analysis of AR, CXXC5, ID1, ID3, PFN2, TET2, BRD4, and p300 in C4-2$^{CON}$ and C4-2$^{ENZ-R}$ cells treated with NEO2734 at the indicated concentrations. ERK2 was used as a loading control; Experiments were repeated twice. **e–g** Effect of the indicated inhibitors on C4-2$^{ENZ-R}$ xenograft tumor volume and weight. Data are represented as means ± s.d. ($n = 10$ replicates/group). Statistical significance was determined by two-way ANOVA for tumor volume and by unpaired two-tailed Student's $t$ tests for tumor weight. **h** Effect of the indicated inhibitors on mouse body weight. Data shown as means ± s.d. ($n = 10$ replicates/group). **i** IHC of AR, CXXC5, ID1, PFN2, Ki-67, and cleaved-caspase3 in tumors treated for 28 days with vehicle, ENZ, or NEO2734. Representative images (scale bar, 100 μm) and quantified data are shown in upper and lower panels, respectively. Data shown as means ± s.d. ($n = 5$ replicates/group). Statistical significance was determined by unpaired two-tailed Student's $t$ tests.

**Lentiviral shRNA infection and cell proliferation assays**. 293T cells were co-transfected with control shRNA, *AR*, *CXXC5*, *TET2*, *ID1*, *ID3*, or *PFN2* gene-specific shRNA (shRNA sequences are provided in Supplementary Data 3) along with packing and envelop plasmids by Lipofectamine 2000 according to the manufacturer's instructions. At 2 days post transfection, virus particles containing shRNA were used to infect PCa cells according to the protocol provided by Sigma-Aldrich. Cells were transduced in culture with a 1:1 mixture of fresh medium and virus supernatant with Polybrene (4 μg/ml final concentration) for 24 h. For cell proliferation analysis, cells infected with sh*AR*, sh*CXXC5*, sh*TET2*, sh*ID1*, sh*ID3*, sh*PFN2*, or control shRNA were seeded in 96-well plates (3000 cells/well) and cultured in medium containing 10% FBS. Cells were fixed at different time points (day 0–5) and cell growth was measured using sulforhodamine B (SRB) assay[63].

**Generation of xenografts, PDXs and organoids, and drug treatment**. Male SCID mice were generated in house and used for animal experiments. All mice were housed under standard conditions with a 12 h light/dark cycle and access to food and water ad libitum and maintained under pathogen-free conditions. The animal study was approved by the Institutional Animal Care and Use Committee (IACUC) at the Mayo Clinic. C4-2$^{ENZ-R}$ cells ($3 \times 10^6$) were mixed with Matrigel (in 50 μl of 1× PBS plus 50 μl of Matrigel (BD Biosciences)) and injected subcutaneously into the right flank of 6-week-old castrated mice. After xenografts reached a size of ~100 mm$^3$ and then animals were randomized into one of five treatment groups ($n = 5$–10 per group) including vehicle (10% DMSO, 40% polyethylene glycol 400, and 50% saline), ENZ (10 mg per kg body weight), CPI637 (10 mg per kg body weight), JQ1 (50 mg per kg body weight), NEO2734 (10 mg per kg body weight) or combination of CPI637 and JQ1. For the ENZ-R PCa PDX study, PDXs previously generated in the laboratory of Liewei W at Mayo Clinic[45] were divided into small pieces (~1 mm$^3$) and injected subcutaneously (s.c.) into 6-week-old castrated mice. After xenografts reached a size of ~100 mm$^3$, animals were randomized into one of five treatment groups ($n = 8$ per group) in a manner similar to the C4-2$^{ENZ-R}$ xenograft study. Treatments were administrated individually 5 days per week by oral gavage and growth in tumor volume was measured in a blinded fashion using digital calipers. Tumor volume was calculated using the following equation: tumor volume = length × width × width × 0.5.

For organoid studies, organoids established from PDXs[64] were embedded in 40 μl of Matrigel and cultured in FBS-free DMEM/F-12 medium supplemented with growth factors EGF (50 ng ml$^{-1}$), FGF2 (5 ng ml$^{-1}$), FGF10 (10 ng ml$^{-1}$), B27 (2%), Prostaglandin E2 (1 μM), SB202190 (10 μM), Y-27632 (10 μM), Nicotinamide (12 mg ml$^{-1}$), N-acetylcysteine (1.25 mM), R-spondin (500 ng ml$^{-1}$), and Noggin (100 ng ml$^{-1}$). The volume of organoids (diameter >5 μm) was measured using Image-Pro and the average of organoid volumes for different treatment were calculated. Cell viability assays for organoids were conducted by plating 2000 organoid cells per well of a collagen-coated 96-well cell culture plate in 100 μl of medium with vehicle (DMSO), ENZ or NEO2734 as indicated. Viable cells were counted using a CellTiter-Glo (Promega) Luminescent Cell Viability Assay Kit.

**Immunohistochemistry (IHC)**. PCa specimens used for IHC were obtained from the Tissue Registry of Mayo Clinic (Rochester, MN) and the Prostate Centre at the University of British Columbia (Vancouver, BC, Canada). The studies were approved by the Institute Review Board of Mayo Clinic. Informed consent was obtained from the study participants. All specimens were de-identified from patient information. FFPE tumor samples from patients, PDXs or C4-2$^{ENZ-R}$ xenograft tumors were deparaffinized, rehydrated and subjected to heat-mediated antigen retrieval. Sections were incubated with 3% $H_2O_2$ for 15 min at room temperature to quench endogenous peroxidase activity. After antigen retrieval using unmasking solution (Vector Labs), slides were blocked with normal goat serum for 1 h and then incubated with primary antibody at 4 °C overnight. IHC analysis of tumor samples was performed using primary antibodies for CXXC5 (dilution 1:500; #16513-1-AP, Proteintech), AR (dilution 1:1000; #ab108341, Abcam), ID1 (dilution 1:1000; #ab66495, Abcam) and PFN2 (1:1000; #LS-C186004-100, LSBio). The sections were then washed three times in 1× PBS and treated for 30 min with biotinylated goat-anti-rabbit IgG secondary antibodies (#BA-9200, Vector Labs).

After washing three times in 1× PBS, sections were incubated with streptavidin-conjugated HRP (#3999S, Cell Signaling Technology). After washing three times in 1× PBS for 5 min each, specific detection was developed with 3,3′-diaminobenzidine (#D4168-50SET, Sigma-Aldrich). Images were acquired using a Leica camera and software. IHC staining was scored on the basis of the 'most common' criteria. Staining score = staining intensity × staining positivity. Staining intensity was graded into four categories: 0, 1, 2, and 3. Specifically, 0 = no staining, 1 = weak staining (staining obvious only at ×400), 2 = medium staining (staining obvious at ×100 but not ×40), and 3 = strong staining (staining obvious at ×40). For staining positivity, 0 = no positive cells, 1 ≤ 10% of positive cells, 2 = 10–50% positive cells, 3 = 51–70% positive cells, 4 ≥ 70% positive cells.

**Western blotting**. Cells were lysed and boiled for 10 min in sample buffer (2% SDS, 10% glycerol, 10% β-Mercaptoethanol, Bromphenol Blue and Tris-HCl, pH 6.8). Equal amounts of protein (50–100 μg) from cell lysate were denatured in sample buffer (Thermo Fisher Scientific), subjected to SDS-polyacrylamide gel electrophoresis, and transferred to nitrocellulose membranes (Bio-Rad). The membranes were immunoblotted with specific primary antibodies, horseradish peroxidase-conjugated secondary antibodies, and visualized by SuperSignal West Pico Stable Peroxide Solution (Thermo Fisher Scientific). Primary antibodies were against AR (dilution 1:1000; #sc-816, Santa Cruz Biotechnology), CXXC5 (dilution 1:1000; #16513-1-AP, Proteintech), CXXC4 (dilution 1:500; #ab105400, Abcam), TET2 (dilution 1:1000; #MABE462, Millipore), TET3 (dilution 1:1000; #ab139311, Abcam), TET1 (dilution 1:1000; #ab191698, Abcam), ID3 (dilution 1:500; #sc-56712, Santa Cruz Biotechnology), PFN2 (dilution 1:1000; #sc-100955, Santa Cruz Biotechnology), BRD4 (dilution 1:1000; #ab128874, Abcam), p300 (dilution 1:1000; #MS-586-PO, Thermo Scientific), ID1 (dilution 1:1000; #ab66495, Abcam), FOXA1 (dilution 1:1000; #ab23738, Abcam), Flag (dilution 1:1000; #F1804, Sigma Aldrich) and V5 (dilution 1:1000; #A190-120A, Bethyl Laboratories) and ERK2 (dilution 1:2000; #sc-1647, Santa Cruz Biotechnology).

**Immunoprecipitation, and protein purification and pulldown assay**. For the His-tag pulldown assay, purified His-tagged CXXC5 or His-tag control lysis were incubated with Flag-AR, V5-TET2 alone, or Flag-AR and V5-TET2 combination in binding buffer containing 10 mM imidazole for 4 h at 4 °C. Ni-NTA beads (Qiagen) for 3 h at 4 °C, washed with wash buffer containing 10 mM imidazole, and eluted with SDS sample buffer. Antibodies for protein purification were Flag (#F1804, Sigma Aldrich) and V5 (#A190-120A, Bethyl Laboratories). The AR, TET2, and its mutant proteins were produced by TNT Quick Coupled Transcription/Translation System (Promega, # L1170) following the manufacturer's instructions.

**Methylated DNA immunoprecipitation (MeDIP)**. Cells were treated with proteinase K and RNase A and genomic DNA was isolated by phenol/chloroform extraction and sonicated. 4 μg of fragmented genomic DNA was immunoprecipitated with antibodies for 5hmC (0.5 μg, #39769, Active Motif) or 5mC (1 μg, #39649, Active Motif) in binding buffer (10 mM Na-Phosphate pH 7.0, 140 mM NaCl, 0.05% Triton X-100) for 4 h. Protein A/G beads were added to each reaction and incubated at 4 °C for 2 h and then washed with IP buffer for three times. The beads were suspended in proteinase K digestion buffer (50 mM Tris pH 8.0, 10 mM EDTA, 0.5% SDS, 200 μg/ml protease K) and incubated at 55 °C for 3 h. Samples were extracted using QIAquick PCR Purification Kit (Cat No. 28106). The immunoprecipitated DNA was quantitatively measured by real-time PCR (qPCR) analysis using Power SYBR Green (#4368708, Thermo Fisher). The primers are same as those used in ChIP-qPCR listed in Supplementary Data 3.

**Analysis of 5hmC levels using dot blot**. For 5hmC detection, genomic DNA samples were denatured and twofold serial dilutions were spotted on a nitrocellulose membrane (#88014, Thermo Fisher). The blotted membrane was UV-cross-linked (70,000 μJ/cm$^2$) and blocked with 5% skimmed milk powder for 1 h at room temperature and incubated with anti-5hmC antibody (dilution 1:1000;

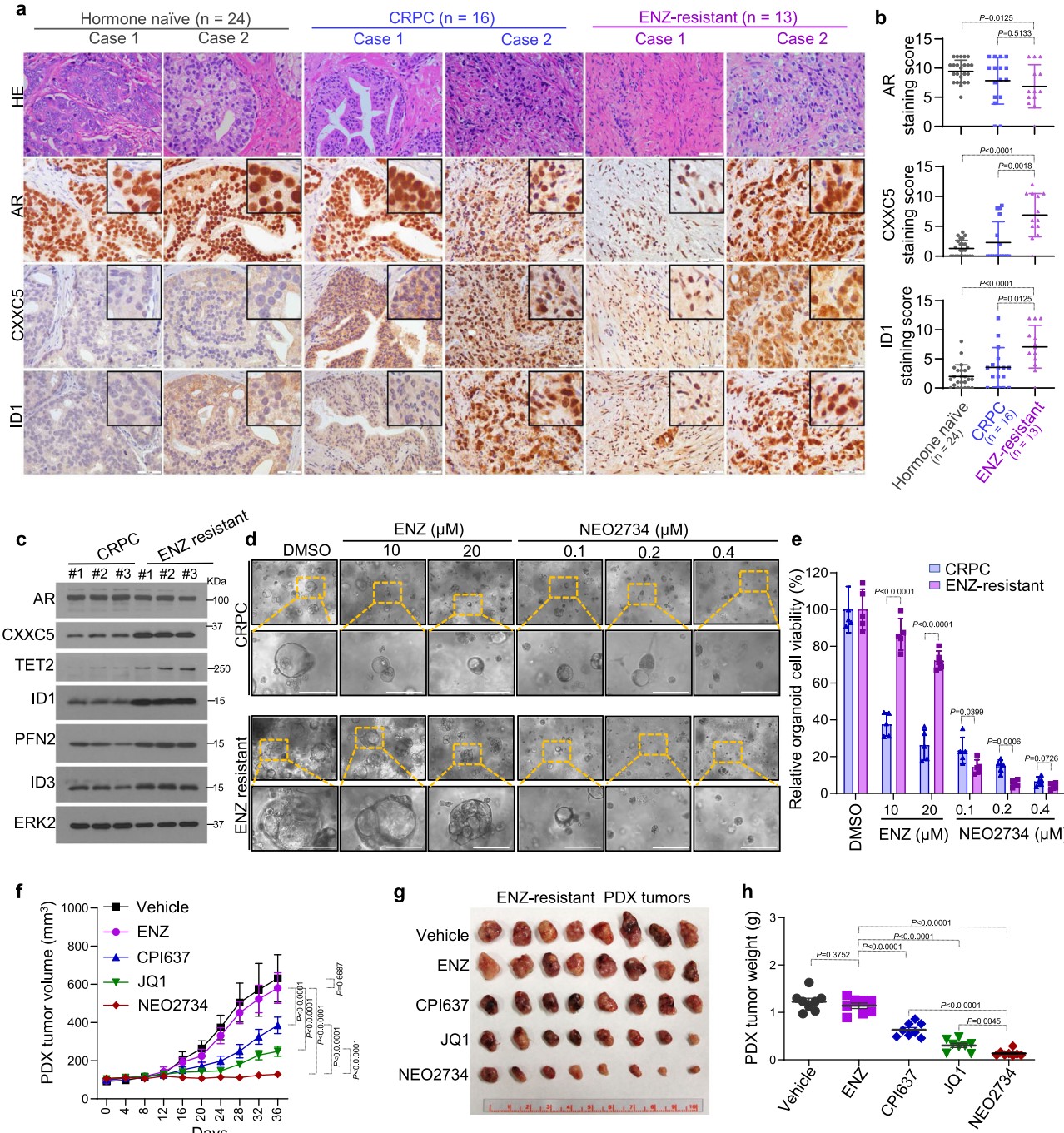

**Fig. 6 Noncanonical AR target expression in ENZ-R CRPC patient samples and therapeutic targeting of noncanonical AR activity in organoids and PDXs.** Representative images for HE staining and IHC of AR, CXXC5, and ID1 in hormone naïve ($n = 24$), CRPC ($n = 16$), and ENZ-resistant ($n = 13$) PCa patient specimens (scale bar, 50 μm) (**a**) and quantification of IHC (**b**). Data are represented as means ± s.d. Statistical significance was determined by unpaired two-tailed Student's $t$ tests. **c** Western blot analysis of AR, CXXC5, TET2, ID1, PFN2, and ID3 in CRPC and ENZ-resistant PCa PDXs (three tumors for each PDX). **d**, **e** Organoids derived from CRPC and ENZ-resistant PDX tumors were treated with the indicated inhibitors for 10 days. Representative images (scale bar, 100 μm) were taken (**d**) and cell viability was determined (**e**). Data shown as ±s.d., ($n = 5$ replicates/group). Statistical significance was determined by unpaired two-tailed Student's $t$ tests. **f**–**h** Effect of the indicated inhibitors on ENZ-resistant PDX tumor volume and weight; Data shown as ±s.d., ($n = 8$ replicates/group). Statistical significance was determined by two-way ANOVA for tumor volume and by unpaired two-tailed Student's $t$ tests for tumor weight.

#39769, Active Motif) at 4 °C overnight. The membrane was subjected to immunoblot analysis using HRP-conjugated IgG secondary antibody (dilution: 1:10000; #7074, Cell Signaling Technology) for 1 h at room temperature. To ensure equal spotting of total DNA on the membrane, the blot was stained with 0.02% methylene blue in 0.3 M sodium acetate (pH 5.2).

**Statistical analysis**. GraphPad Prism 7 was used for statistical analyses with the qPCR, cell proliferation analysis, tumor growth analysis, and IHC quantification data. $P$ values from unpaired two-tailed Student's $t$ tests were used for comparisons between two groups and one-way ANOVA with Bonferroni's post hoc test was used for multiple comparisons. For treatment time course experiments, two-way

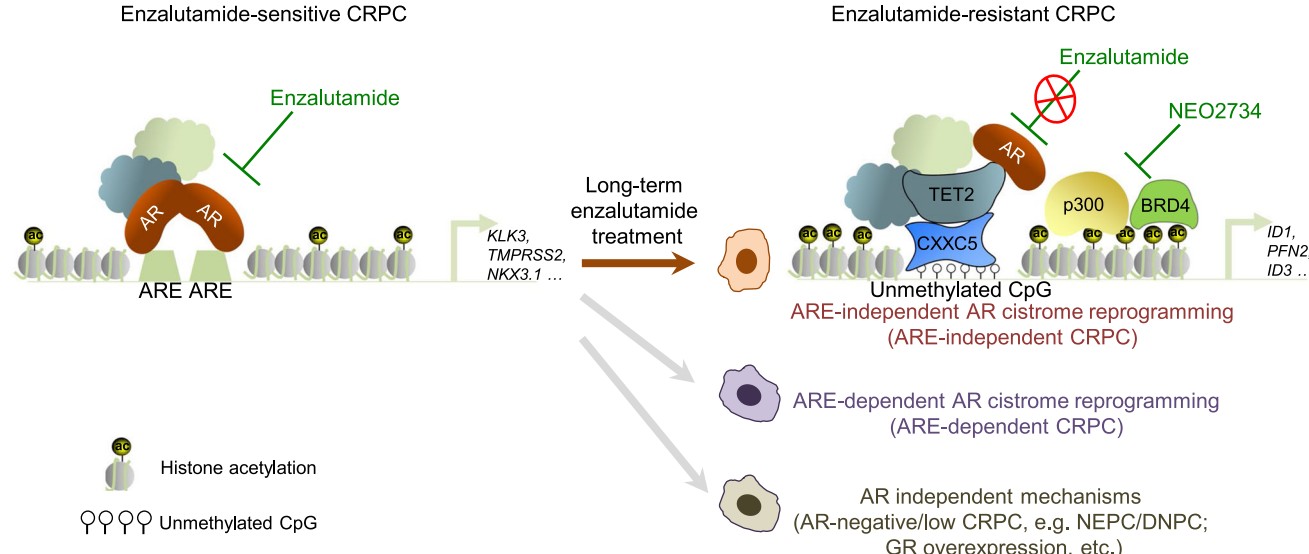

**Fig. 7 A hypothetical model deciphering the mechanisms of EZN-sensitive and ENZ-resistant CRPC.** AR androgen receptor, ARE androgen response element, ac acetylation, CRPC castration-resistant prostate cancer, NEPC neuroendocrine prostate cancer, DNPC double-negative (AR-null/neuroendocrine-null) prostate cancers, GR glucocorticoid receptor, CXXC5 CXXC Finger Protein 5, TET2 Tet Methylcytosine Dioxygenase 2, BRD4 Bromodomain-containing protein 4, p300 Histone acetyltransferase p300.

ANOVA followed by post hoc test was applied. Statistical analysis is indicated in each figure legend. $P$ value $< 0.05$ was considered significant.

**Reporting summary**. Further information on research design is available in the Nature Research Reporting Summary linked to this article.

## Data availability

The RNA-seq and ChIP-seq data are deposited in the National Center for Biotechnology Information (NCBI) Gene Expression Omnibus (GEO) database with the accession code GSE136130: All relevant data are available from the authors. Source data are provided with this paper.

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

## Acknowledgements

This work was supported in part by grants from the National Institutes of Health (CA134514, CA130908, CA193239 and CA203849 to H.H.), the Mayo Clinic Foundation (to H.H.), the Canadian Institutes of Health Research operating grants (141635, 144159, and 153081 to Y.W.), and Terry Fox Research Institute program project (1062 to Y.W.). Y.H. is supported by Mayo Edward C. Kendall Fellowship. The authors would like to thank Epigene Therapeutics Inc for providing the compound NEO2734.

## Author contributions

H.H., Y.H., L.S. designed the study. Y.H. generated reagents and performed experiments, data collection and analysis. T.W., Z.Y., J.J.O., H.S., Liguo W. performed data mining and ChIP-seq and RNA-seq bioinformatics analysis. Y.H., D.L., L.F., R.J.K., R.J., Y.W. acquired patient tissue samples and performed IHC staining and scoring. Liewei W., Y.W. generated PDXs. H.H., Y.H., L.S., J.J.O. wrote the manuscript.

## Competing interests

The authors declare no competing interests.
