## [Peer Review File · Nature Communications]

Reviewers' Comments:

Reviewer #1:

Remarks to the Author:

The manuscript titled, "An unorthodox AR addiction drives enzalutamide resistance in prostate cancer," describes studies examining alterations in androgen receptor (AR) binding sites in enzalutamide resistant prostate cancer cells, with associations of these binding sites with other chromatin modulators. These studies nominate the involvement of CXXC5 and TET2 in recruiting AR to novel sites in the genome leading to induction of a new transcriptional program that confers enzalutamide resistance. This topic, mechanisms of resistance to second generation anti-androgen therapies such as enzalutamide, is a key area of interest in the prostate cancer field. The studies hold significant merit. I have the following constructive criticisms that I hope can help further improve the manuscript:

- 1) Rescue experiments would be helpful to understand the mechanism by which CXXC5 and TET2 are involved in enzalutamide resistance. For example, as an addition to the loss of function studies, does rescue with WT vs. catalytically inactive TET2, or CXXC5 mutants/deletion constructs incapable of binding TET2, reverse the knockdown phenotype?
- 2) If the catalytic activity of TET2 is implicated, then is there a change in DNA methylation or hydroxymethylation at the target genes in the control and enzalutamide resistant cells, and is this modulated by TET2 loss?
- 3) The therapeutic strategies with the CBP/P300 inhibitors are not clearly tied to the mechanism that is proposed. These drugs could have many effects not related to the proposed mechanisms. To demonstrate a connection, they would need to see if clamping CXXC5 and TET2 expression leads to attenuated response to the drug.
- 4) AR has been shown to bind to enhancer regions in many prostate cancer cell systems including both castration sensitive and resistant. It would be good to examine the association of AR with putative enhancers in the models. The H3K27ac combined with other enhancer marks including H3K4 methylation or MED1, may be useful here. Use of publicly available data for the control cell lines could be leveraged.
- 5) Figure 2: The association of gained AR binding sites with CpG islands could simply be due to the increased association of these gained AR binding sites with promoter regions, since a large fraction of promoters contain CpG islands and high GC content. This is further corroborated by the fact that 50% of the CpG island loci in the AR gained sites are associated with promoters (this is not very different than the fraction of all promoters that contain CpG islands, estimated to be about 50-60% in previous studies).
- 6) Figure 6: it would be helpful to evaluate TET2 expression in the patient cases.

Minor Comments:

- 1) Figure 1G: this is somewhat deceiving since each arc along the circumference appears to be the same length. I am not sure this Circos plot format is the best way to present these associations.
- 2) The phrase "hormone-naïve" to describe LNCaP cells, and metastatic CRPC to describe C4-2B cells is not entirely appropriate. C4-2B is a castration resistant sub-line of LNCaP cells. LNCaP is itself from a lymph node metastasis.

Reviewer #2:

Remarks to the Author:

The study is focused on understanding resistance to enzalutamide in prostate cancer cells. The major conclusion is that treatment of prostate cancer cells with Enz results in a shift in occupancy of the Androgen Receptor to non-canonical sites and to regulation of a distinct gene profile. This is a conclusion that would be expected from the extensive previous literature on Enz resistance in prostate cancer cells.

The new findings link the gained sites -- which they call ARBS-gained - are enriched in CpG island and overlap with CXXC5. Downstream targets of CXXC5 are changes in the ENZ cell lines.

The results are interesting but not compelling. In many of the data figures, the differences between the groups may be statistically significant but they are subtle. The drug treatment studies in particular are subtle and not compelling.

In addition the study is mostly descriptive in nature, and mostly limited to analyses of cell lines with limited extension to human patient data (a few cases in Fig. 6)

The study could be streamlined to better emphasize the novel aspects and to move some of the detail to the supplementary data.

Specific comments:

line 65 - this reviewer does not think it is surprising that AR limits proliferation of these cells.

figure 1 - perhaps most of this figure would be suited for supplementary materials

line 117 - it would be helpful to know if CXXC4 or 5 are relevant for ENZ resistant patients.

line 143 - up regulation of neurogenesis genes is not indicative of a neuro-related phenotype.

line 168 - what is the clinical endpoint Fig. 3f? This is data from the SU2C cohort.

Fig. 4 - most of the changes are modest

In Fig. 5 - some of the panels have modest differences; in the IHC in panel I, the NEO2734 - looks like it is not stained as well as the others overall

Reviewer #3:

Remarks to the Author:

The next-generation anti-AR drugs such as enzalutamide (ENZ) extend the lives of CRPC patients but tumors rapidly develop ENZ resistance. Many mechanisms have been reported to mediate ENZ resistance. Here, by comparing the genome-wide ChIP-seq and RNA-seq data, authors identify, in ENZ-resistant PCa cell line models (mainly C4-2 cells chronically treated with ENZ), a unique set of ARBS that lack the canonical DNA-binding elements of AR (i.e., AREs) and FOXA1. Termed ARBS-gained sites, authors found that they overlap significantly with CXXC5 binding sites. Of interest, the ARBS-gained loci are enriched in H3K27ac, a histone modification mark of enhancers. This latter observation led to new findings that ENZ-resistant PCa models are sensitive to dual BET-CBP/p300 inhibitor NEO2734. Overall, authors identify a novel ENZ-resistant mechanism driven by AR-dependent transcription of non-canonical targets including ID1, ID3 and PFN2. Their studies also led to the proof-of-principle that clinically, ENZ-resistant AR+ CRPC might be susceptible to dual inhibition of BET and CBP/p300 signaling.

Main points:

1. There are multiple mechanisms that mediate ENZ resistance including de novo resistance. Authors are AR-centric and ignore a large body of literature on ARneg/low (or AR activity-low or AR-indifferent) PCa cells/clones that pre-exist in untreated primary tumors, which tend to become accentuated in metastatic foci of the mCRPC patients (e.g., Alumkai JJ et al., PNAS, 117:12315-12323, 2020; Li Q et al., Nat. Commun., 9:3600, 2018; Spratt DE et al., Cancer Res, 25:6721-6730, 2019; Labrecque MP et al., JCI, 129:4492, 2019). Importantly, the 'unorthodox' AR genomic binding at the unmethylated CpGi was discovered in the cell line model C4-2, and it's unknown whether this noncanonical 'AR genomic signature' actually exists in the ENZ-treated CRPC in patients, as it is well known that AR targets in castration-resistant PCa cell line models can be quite different from castration-resistant tumors in patients. Authors should try to interrogate ENZ-treated (or ENZ-resistant) patients' tumor transcriptomic data for the unorthodox AR genomic signature they reported.
2. About 20% of ARBS-gained loci overlapped CpGi regions (p5, top), which should correspond to ~1,380 ARBS-gained sites. More than 50% (~54%; Fig. 2d) of ARBS-gained CpGi sites were

located at the promoter region', which should be ~750 sites instead of '(approximately 1,000 sites)'.

3. Of the ~750 ARBS-gained CpGi loci, 'Over 50%.... genes were significantly upregulated in C4-2ENZ-R cells.....', but where is the list of these genes? If it's Table S1, then authors need to provide a legend to explain the scores presented, the exact genes upregulated, and statistical parameters.

4. The C4-2ENZ-R cells upregulate both CXXC5 and TET2 and show enriched CXXC5 and TET2 ChIP binding signals at the ARBS-gained CpGi (Fig. 2l, m). The authors seemed to be thinking that it's the upregulated CXXC5 that mediated AR contact with the unmethylated CpGi, but what's the exact role of TET2 in this context?

5. What could be the potential mechanisms for NEO2734-induced degradation of AR, CXX5 and ID proteins (Fig. 5c)?

Authors' Response to Reviewers' Comments on MS# NCOMMS-20-22879

REVIEWER COMMENTS

Reviewer #1 (Remarks to the Author):

The manuscript titled, "An unorthodox AR addiction drives enzalutamide resistance in prostate cancer," describes studies examining alterations in androgen receptor (AR) binding sites in enzalutamide resistant prostate cancer cells, with associations of these binding sites with other chromatin modulators. These studies nominate the involvement of CXXC5 and TET2 in recruiting AR to novel sites in the genome leading to induction of a new transcriptional program that confers enzalutamide resistance. This topic, mechanisms of resistance to second generation anti-androgen therapies such as enzalutamide, is a key area of interest in the prostate cancer field. The studies hold significant merit. I have the following constructive criticisms that I hope can help further improve the manuscript:

Reply: We very much thank the Reviewer for the positivity and valuable advices for improvement of the manuscript.

1) Rescue experiments would be helpful to understand the mechanism by which CXXC5 and TET2 are involved in enzalutamide resistance. For example, as an addition to the loss of function studies, does rescue with WT vs. catalytically inactive TET2, or CXXC5 mutants/deletion constructs incapable of binding TET2, reverse the knockdown phenotype?

Reply: To explore whether the catalytic activity of TET2 involves in enzalutamide resistance, we generated a catalytically inactive TET2 mutant, TET2-H1881R. We performed 5hmC dot blot analysis and confirmed that it is an enzymatic inactivation mutant (Fig. 4j, left).

Co-IP analysis (Fig. 4j, right) and in vitro protein binding assay (Fig. 4k) showed that both TET2-WT and TET2-H1881R enabled to mediate CXXC5 binding of AR. Rescue experiments showed that restored expression of both TET2-H1881R and TET2-WT invariably rescued TET2 knockdown-induced downregulation of noncanonical AR genes (ID1, ID3 and PFN2) and inhibition of ENZ-R cell growth (Fig. 4l-n). These results indicate the catalytic activity of TET2 is not involved in enzalutamide resistance.

Furthermore, restored expression of CXXC5-WT but not the TET2 binding-deficient mutant CXXC5(1-250) rescued CXXC5 knockdown-induced downregulation of noncanonical AR target genes and inhibition of ENZ-R cell growth (Supplementary Fig. 5a-d), suggesting that the binding of CXXC5 with TET2 is critical for CXXC5-mediated enzalutamide resistance.

2) If the catalytic activity of TET2 is implicated, then is there a change in DNA methylation or hydroxymethylation at the target genes in the control and enzalutamide resistant cells, and is this modulated by TET2 loss?

Reply: As discussed above, our new data showed that the catalytic activity of TET2 is not involved in enzalutamide resistance. Consistent with these results, results from methylated DNA immunoprecipitation (MeDIP) assay showed that the DNA 5mC/5hmC levels at the noncanonical AR target loci, including ID1, PFN2 and ID3, were hardly detectable compared to the positive control locus and unaffected by knockdown of TET2 or elimination of TET2 catalytic activity (Supplementary Fig. 5e).

3) The therapeutic strategies with the CBP/P300 inhibitors are not clearly tied to the mechanism that is proposed. These drugs could have many effects not related to the proposed mechanisms. To demonstrate a connection, they would need to see if clamping CXXC5 and TET2 expression leads to attenuated response to the drug.

Reply: We appreciate the great point. The unbiased drug sensitivity survey revealed that enzalutamide resistant cells were hypersensitive to CBP/p300 inhibitor and BET bromodomain inhibitor (Fig. 5a), which prompted us to test the effect of the newly developed BET-CBP/p300 dual inhibitor NEO2734 in this study. Our studies suggest that CXXC5 and TET2 play important roles in the growth of enzalutamide-resistant cells (Fig. 2h, i); however CXXC5 but not TET2 was downregulated by treatment of NEO2734 (Fig. 5d). Therefore, we chose to primarily focus on examining whether CXXC5 expression level is critical for cell response to NEO2734.

We found that forced overexpression of CXXC5 not only attenuated NEO2734-induced downregulation of ID1, ID3 and PFN2 genes, but also restored TET2 and AR binding at these gene loci (Supplementary Fig. 6i-k). Overexpression of CXXC5 also attenuated the NEO2734-induced inhibition of C4-2-ENZ-R cell growth (Supplementary Fig. 6l). These data indicate that clamping CXXC5 expression and the enhanced CXXC5/TET2 complex lead to attenuated response to the BET-CBP/p300 dual inhibitor NEO2734.

4) AR has been shown to bind to enhancer regions in many prostate cancer cell systems including both castration sensitive and resistant. It would be good to examine the association of AR with putative enhancers in the models. The H3K27ac combined with other enhancer marks including H3K4 methylation or MED1, may be useful here. Use of publicly available data for the control cell lines could be leveraged.

Reply: Thanks for the comments. By analyzing the publicly available ChIP-seq data of H3K4me1 (an enhancer histone mark) in ENZ-resistant LNCaP-AR cells (GSE103449), we identified a total of 1,065,164 putative enhancers (Supplementary Fig.2b). Importantly, we found that ARBS-gained, ARBS-lost and ARBS-NSA loci have substantial overlaps with these putative enhancers (Supplementary Fig.2b). Notably, we found that only ARBS-gained loci at the putative enhancer regions, but not those of ARBS-NSA and ARBS-lost loci, exhibited increased enrichment of H3K27ac in C4-2 ENZ-R cells compared to C4-2 control cells (Supplementary Fig. 2b). These new data suggest that increased enrichment of H3K27ac at the ARBS-gained loci in ENZ-resistant cells is unlikely promoter-specific.

5) Figure 2: The association of gained AR binding sites with CpG islands could simply be due to the increased association of these gained AR binding sites with promoter regions, since a large fraction of promoters contain CpG islands and high GC content. This is further corroborated by the fact that 50% of the CpG island loci in the AR gained sites are associated with promoters (this is not very different than the fraction of all promoters that contain CpG islands, estimated to be about 50-60% in previous studies).

Reply: This is an excellent point. To address this, we examined the percentage of CpG-positive loci among the total ARBS-gain loci in the promoter regions. We found that about 82% of ARBS-gained loci in the promoter regions are CpG island-positive (Fig. 1h and Supplementary Table 1), which is much higher than the estimated 50-60% of all promoters that contain CpG islands (*Genes Dev. 2011 May 15, 25(10):1010-1022.*). These data rule out the possibility that the association of gained AR binding sites with CpG islands is simply due to the increased association of these gained AR binding sites with promoter regions.

6) Figure 6: it would be helpful to evaluate TET2 expression in the patient cases.

Reply: Thanks for the great comment. The standard practice of IHC in our lab is that the criterion for a given antibody to be suitable for IHC is that only one band is shown up in the western blot. Compared to other antibodies we used in our IHC studies (anti-AR, anti-CXXC5, anti-ID and anti-PFN2), there are multiple bands on the western blots of two anti-TET2 antibodies we tested, especially in TET2 knockdown cells (see data below). Therefore, we are not comfortable to use these two anti-TET2 antibodies for IHC in patient tissues. While we are still searching for suitable TET2 antibodies for IHC, we need to defer this experiment to the future because of this technical limitation.

Minor Comments:

1) Figure 1G: this is somewhat deceiving since each arc along the circumference appears to be the same length. I am not sure this Circos plot format is the best way to present these associations.

Reply: We agree and we have changed the way to present this data in Fig. 1c.

2) The phrase “hormone-naïve” to describe LNCaP cells, and metastatic CRPC to describe C4-2B cells is not entirely appropriate. C4-2B is a castration resistant sub-line of LNCaP cells. LNCaP is itself from a lymph node metastasis.

Reply: Thanks for the comment. We have described LNCaP and C4-2B cell lines as suggested by the Reviewer (see page 6).

Reviewer #2 (Remarks to the Author):

The study is focused on understanding resistance to enzalutamide in prostate cancer cells. The major conclusion is that treatment of prostate cancer cells with Enz results in a shift in occupancy of the Androgen Receptor to non-canonical sites and to regulation of a distinct gene profile. This is a conclusion that would be expected from the extensive previous literature on Enz resistance in prostate cancer cells.

The new findings link the gained sites -- which they call ARBS-gained - are enriched in CpG

island and overlap with CXXC5. Downstream targets of CXXC5 are changes in the ENZ cell lines.

The results are interesting but not compelling. In many of the data figures, the differences between the groups may be statistically significant but they are subtle. The drug treatment studies in particular are subtle and not compelling.

In addition the study is mostly descriptive in nature, and mostly limited to analyses of cell lines with limited extension to human patient data (a few cases in Fig. 6)

The study could be streamlined to better emphasize the novel aspects and to move some of the detail to the supplementary data.

Reply: Thank the Reviewer for the insightful comments and advices. As suggested by the Reviewer, we have moved some of the detailed results such as establishment and characterization of different ENZ-resistant cell lines in culture and in mice as well as AR cellular localization and knockdown data to supplementary data (Supplementary Fig. 1 and 2) and leaves the novel findings in the revised Fig. 1 by focusing on how we identified ARBS-gain site in enzalutamide resistant cells and what is the major genomic features of the ARBS-gained sites.

Specific comments:

line 65 - this reviewer does not think it is surprising that AR limits proliferation of these cells.

Reply: Thanks for the comment. We have removed the wording "To our surprise".

figure 1 - perhaps most of this figure would be suited for supplementary materials

Reply: As discussed above, we have moved some of the detailed results such as establishment and characterization of different ENZ-resistant cell lines in culture and in mice as well as AR cellular localization and knockdown data to supplementary data (Supplementary Fig. 1 and 2) and leaves the novel findings in the revised Fig. 1 by focusing on how we identified ARBS-gain site in enzalutamide resistant cells and what is the major genomic features of the ARBS-gained sites.

line 117 - it would be helpful to know if CXXC4 or 5 are relevant for ENZ resistant patients.

Reply: Great point. We performed meta-analysis of RNA-seq data in the SU2C database. We found that high expression of CXXC5 associated with poor overall survival of patients treated with AR signaling inhibitors although the P value of the association was slightly greater than 0.05 (Fig. 3d). This could be explained in the context that the impact of CXXC5 on therapy resistance may greatly rely on the functions of its downstream effectors. Indeed, high expression scores of CXXC5-regulated noncanonical AR signature genes (n=226), including ID1, PFN2 and ID3 (Fig. 3a; Supplementary Table 2), significantly associated with unfavorable outcome of AR signaling inhibitors in CRPC patients (Fig. 3e-g). Particularly, high expression of ID1 or PFN2 gene alone also significantly associated with poor outcome of AR-targeted therapy in these patients (Fig. 3h). CXXC4 was excluded from analysis since its expression was much lower compared with CXXC5 in these patients (Supplementary Fig. 4d).

line 143 - up regulation of neurogenesis genes is not indicative of a neuro-related phenotype.

Reply: We agree and have removed the phrase of “neuro-related phenotype”.

line 168 - what is the clinical endpoint Fig. 3f? This is data from the SU2C cohort.

Reply: Yes, this data is from the SU2C cohort and the clinical endpoint is “overall survival”, which has been specified in Fig. 3d, e and h.

Fig. 4 - most of the changes are modest

Reply: We thank Reviewer for the comments. We have moved the drug treatment data to supplementary information (Supplementary Fig. 4e, f). Instead, more mechanistic data such as ChIP analysis (Fig. 4i) and rescue experiments (Fig. 4j-n) have been added to the revised Fig. 4.

In Fig. 5 - some of the panels have modest differences; in the IHC in panel I, the NEO2734 - looks like it is not stained as well as the others overall

Reply: We repeated the IHC experiments in Figure 5 and new data are shown in the revised Fig. 5i.

Reviewer #3 (Remarks to the Author):

The next-generation anti-AR drugs such as enzalutamide (ENZ) extend the lives of CRPC patients but tumors rapidly develop ENZ resistance. Many mechanisms have been reported to mediate ENZ resistance. Here, by comparing the genome-wide ChIP-seq and RNA-seq data, authors identify, in ENZ-resistant PCa cell line models (mainly C4-2 cells chronically treated with ENZ), a unique set of ARBS that lack the canonical DNA-binding elements of AR (i.e., AREs) and FOXA1. Termed ARBS-gained sites, authors found that they overlap significantly with CXXC5 binding sites. Of interest, the ARBS-gained loci are enriched in H3K27ac, a histone modification mark of enhancers. This latter observation led to new findings that ENZ-resistant PCa models are sensitive to dual BET-CBP/p300 inhibitor NEO2734. Overall, authors identify a novel ENZ-resistant mechanism driven by AR-dependent transcription of non-canonical targets including ID1, ID3 and PFN2. Their studies also led to the proof-of-principle that clinically, ENZ-resistant AR+ CRPC might be susceptible to dual inhibition of BET and CBP/p300 signaling.

Reply: We thank the Reviewer for the positivity and valuable comments for improvement.

Main points:

1. There are multiple mechanisms that mediate ENZ resistance including de novo resistance. Authors are AR-centric and ignore a large body of literature on ARneg/low (or AR activity-low or AR-indifferent) PCa cells/clones that pre-exist in untreated primary tumors, which tend to become accentuated in metastatic foci of the mCRPC patients (e.g., Alumkai JJ et al., PNAS, 117:12315-12323, 2020; Li Q et al., Nat. Commun., 9:3600, 2018; Spratt DE et al., Cancer Res, 25:6721-6730, 2019; Labrecque MP et al., JCI, 129:4492, 2019). Importantly, the ‘unorthodox’ AR genomic binding at the unmethylated CpG was discovered in the cell line model C4-2, and it’s unknown whether this noncanonical ‘AR genomic signature’ actually exists in the ENZ-treated CRPC in patients, as it is well known that AR targets in castration-resistant PCa cell line models can be quite different from castration-resistant tumors in patients. Authors should try to interrogate ENZ-treated (or ENZ-resistant) patients’ tumor transcriptomic data for the unorthodox AR genomic signature they reported.

Reply: Thank the Reviewer for the great points. First, we have discussed the importance of Arneg/low PCa cells for mCRPC progression in the second paragraph of the Introduction section and all the references the Reviewer mentioned have been cited.

Second, we performed meta-analysis of RNA-seq data in the SU2C database. We found that high expression of CXXC5 associated with poor overall survival of patients treated with AR signaling inhibitors (including enzalutamide and abiraterone) although the P value of the association was slightly greater than 0.05 (Fig. 3d). This could be explained in the context that the impact of CXXC5 on therapy resistance may greatly rely on the functions of its downstream effectors. Indeed, high expression scores of CXXC5-regulated noncanonical AR signature genes (n=226), including ID1, PFN2 and ID3 (Fig. 3a; Supplementary Table 2), significantly associated with unfavorable outcome of AR signaling inhibitor therapy in CRPC patients (Fig. 3e-g). Particularly, high expression of ID1 or PFN2 gene alone also significantly associated with poor outcome of AR signaling inhibitor therapy in these patients (Fig. 3h).

2. About 20% of ARBS-gained loc overlapped CpGi regions (p5, top), which should correspond to ~1,380 ARBS-gained sites. 'More than 50% (~54%; Fig. 2d) of ARBS-gained CpGi sites were located at the promoter region', which should be ~750 sites instead of '(approximately 1,000 sites)'.

Reply: Thanks for the comments. As suggested by the Reviewer, we have provided the exact number of ARBS-gained loci overlapped with CpGi regions and the exact number of the ARBS-gained CpGi sites located in the promoter regions in the revised manuscript (see page 5).

3. Of the ~750 ARBS-gained CpGi loci, 'Over 50%.... genes were significantly upregulated in C4-2ENZ-R cells.....', but where is the list of these genes? If it's Table S1, then authors need to provide a legend to explain the scores presented, the exact genes upregulated, and statistical parameters.

Reply: Thanks for the comments. The genes which associated with ARBS-gained CpGi loci are listed in Supplementary Table 2. We performed gene differential expression analysis using edgeR (version 3.6.8), and the scores or the Log(fold change) of differential expression of these genes in C4-2^{ENZ-R} cells compared to control cells are included. Specifically, we found that 226 genes were associated with ARBS-gained CpGi regions and they were significantly upregulated (FDR < 0.05) in C4-2^{ENZ-R} cells compared to control cells. These data are shown in Fig. 3a and Supplementary Table 2.

4. The C4-2ENZ-R cells upregulate both CXXC5 and TET2 and show enriched CXXC5 and TET2 ChIP binding signals at the ARBS-gained CpGi (Fig. 2l, m). The authors seemed to be thinking that it's the upregulated CXXC5 that mediated AR contact with the unmethylated CpGi, but what's the exact role of TET2 in this context?

Reply: Excellent point. In vitro protein binding assay showed that CXXC5 physically interacts with TET2. Notably, no CXXC5-AR interaction was detected under similar conditions, but CXXC5 bound to AR after adding TET2 in the experiment system (Fig. 2d, right). Furthermore, ChIP analysis showed that knockdown of TET2 had no obvious effect on CXXC5 occupancy at ID1, ID3 and PFN2 gene loci (Fig. 4i, left), but it decreased AR occupation at these genes loci (Fig. 4i, right). These data indicate that TET2 plays important role in mediating the interaction between CXXC5 and AR and facilitating CXXC5-promoted recruitment of AR into ARBS-gained CpG loci (Fig. 7, right).

5. What could be the potential mechanisms for NEO2734-induced degradation of AR, CXX5 and ID proteins (Fig. 5c)?

Reply: Thanks for the comments. We performed new experiments and demonstrated that NEO2734 treatment not only decreased H3K27ac enrichment and BRD4 and p300 occupancy at CXXC5 target gene loci, but also downregulated mRNA expression of these genes in C4-2 ENZ-R cells (Supplementary Fig. 6g, h). Thus, we mechanistically demonstrated that NEO2734 downregulated AR, CXX5 and ID1 expression at the transcription level by decreasing H3K27ac level and BRD4 and p300 binding at the genomic loci of these genes.

Reviewers' Comments:

Reviewer #1:

Remarks to the Author:

The authors have sufficiently addressed my concerns.

Reviewer #2:

Remarks to the Author:

While the revised manuscript is certainly improved by the revisions, this reviewer does not think that the work, overall, is a significant advance in a crowded field. I also remain concerned about the limited connection to human prostate cancer. The Kaplan Meier results are quite modest and the IHC of patients looks at a very small sample size.

Thus, while the paper is improved, in my opinion it would be better to be published in a more specialized journal.

Reviewer #3:

Remarks to the Author:

In this revised manuscript, authors have addressed most of my earlier questions. There are only some minor language-related issues. For example, in Discussion (page 15, top paragraph), "....., we identify a subset of ARE-independent AR target genes including ID1, PFN2 and ID1 in ENZ-resistance cells." should be "... including ID1, PFN2 and ID3 in ENZ-resistant cells."

Authors' Response to the Reviewers' Comments on NCOMMS-20-22879A

Reviewer #1 (Remarks to the Author):

The authors have sufficiently addressed my concerns.

Reply: We thank the Reviewer for the positivity and enthusiasm about our manuscript.

Reviewer #2 (Remarks to the Author):

While the revised manuscript is certainly improved by the revisions, this reviewer does not think that the work, overall, is a significant advance in a crowded field. I also remain concerned about the limited connection to human prostate cancer. The Kaplan Meier results are quite modest and the IHC of patients looks at a very small sample size. Thus, while the paper is improved, in my opinion it would be better to be published in a more specialized journal.

Reply: We thank the Reviewer for the recognition of the improvement of our revised manuscript. We agree that the Kaplan Meier results are quite modest and the IHC of patients looks at a very small sample size. We also have to admit that it is not uncommon in the field that most of institutions are unable to access to a large number of cases of enzalutamide-resistant CRPC and that our group at Mayo Clinic and the colleagues at the Vancouver Prostate Centre are not exceptional. Nevertheless, we thank the Reviewer for raising this great point and therefore, we acknowledge in the first paragraph of Discussion section that one limitation of our current study is the small sample size and that the significance of our findings can be improved by the validation studies in larger cohorts.

Reviewer #3 (Remarks to the Author):

In this revised manuscript, authors have addressed most of my earlier questions. There are only some minor language-related issues. For example, in Discussion (page 15, top paragraph), "....., we identify a subset of ARE-independent AR target genes including ID1, PFN2 and ID1 in ENZ-resistance cells." should be "... including ID1, PFN2 and ID3 in ENZ-resistant cells."

Reply: We thank the Reviewer for the positivity and enthusiasm about our manuscript. The language related issue/typo has been corrected.